# Improving Subgraph-GNNs via Edge-Level Ego-Network Encodings

**Nurudin Alvarez-Gonzalez** *nuralgon@gmail.com*
*Universitat Pompeu Fabra*

**Andreas Kaltenbrunner** *kaltenbrunner@gmail.com*
*Universitat Oberta de Catalunya*
*ISI Foundation Turin*

**Vicenç Gómez** *vicen.gomez@upf.edu*
*Universitat Pompeu Fabra*

**Reviewed on OpenReview:** *https://openreview.net/forum?id=NOScOKYOAH*

## Abstract

We present a novel edge-level ego-network encoding for learning on graphs that can boost Message Passing Graph Neural Networks (MP-GNNs) by providing additional node and edge features or extending message-passing formats. The proposed encoding is sufficient to distinguish Strongly Regular Graphs, a family of challenging 3-WL equivalent graphs. We show theoretically that such encoding is more expressive than node-based sub-graph MP-GNNs. In an empirical evaluation on four benchmarks with 10 graph datasets, our results match or improve previous baselines on expressivity, graph classification, graph regression, and proximity tasks—while reducing memory usage by 18.1x in certain real-world settings.

## 1 Introduction

Neural graph architectures are the current standard for learning from graph data. These methods automatically learn representations of nodes, edges, or graphs in a data-driven and end-to-end way. Message Passing Graph Neural Networks (MP-GNNs) are the most common model for learning on graphs. MP-GNNs process the input graph (data) with a computational graph (model) to learn useful representations through message passing between direct neighbors in the input graph. This framework facilitates the theoretical analysis of MP-GNNs (e.g., in terms of their expressive power (Xu et al., 2019), or characterizing issues such as *over-squashing* (Alon & Yahav, 2021) or *over-smoothing* (Oono & Suzuki, 2022)).

The idea of decoupling the input graph from the computational graph is the basis of leading-edge learning approaches, such as Sub-graph GNNs (Zhao et al., 2022; Abboud et al., 2022; Frasca et al., 2022; Mitton & Murray-Smith, 2023), perturbation methods (Papp et al., 2021; Dwivedi et al., 2022), or Graph Transformers (Yun et al., 2019; Ying et al., 2021; Rampášek et al., 2022; Kim et al., 2022). These methods extend the message-passing mechanism to more general structures induced by the graph, beyond direct neighbors—for example between the nearest neighbours of a node at a given depth (ego-networks). This flexibility extends the expressive power of MP-GNNs, at the cost of an increased computational footprint and a departure of the inductive bias contained in the input graph, which must be learned again.

In this work, we present an alternative to previous *pure learning* approaches. Rather than learning on sub-graphs, we introduce a systematic procedure to generate a pool of structural features (an encoding) which can subsequently be integrated into MP-GNNs. Similar approaches have been proposed recently (Bouritsas et al., 2023; Alvarez-Gonzalez et al., 2022). Crucially, our proposed features capture information at the edge-level, including signals contained in the two ego-networks of adjacent nodes in the input graph. We call this encoding ELENE, for **E**dge-**L**evel **E**go-**N**etwork **E**ncodings. The benefits of such a representation are diverse:

The encodings are interpretable and amenable for theoretical analysis, they are efficiently computable as a pre-processing step, and finally, they reach comparable performance with state-of-the-art learning methods.

As an illustrative example, consider **S**trongly **R**egular **G**raphs (SRGs). They are known to be *indistinguishable* by node-based sub-graph GNNs (Balcilar et al., 2021; Morris et al., 2023; Papp & Wattenhofer, 2022; Zhao et al., 2022; Frasca et al., 2022), as exemplified by the non-isomorphic $4 \times 4$ Rook and Shrikhande graphs in Fig. 1. We theoretically show that ELENE is as expressive as node-only sub-graph GNNs, and expressive enough to differentiate certain classes of SRGs like those in Fig. 1.

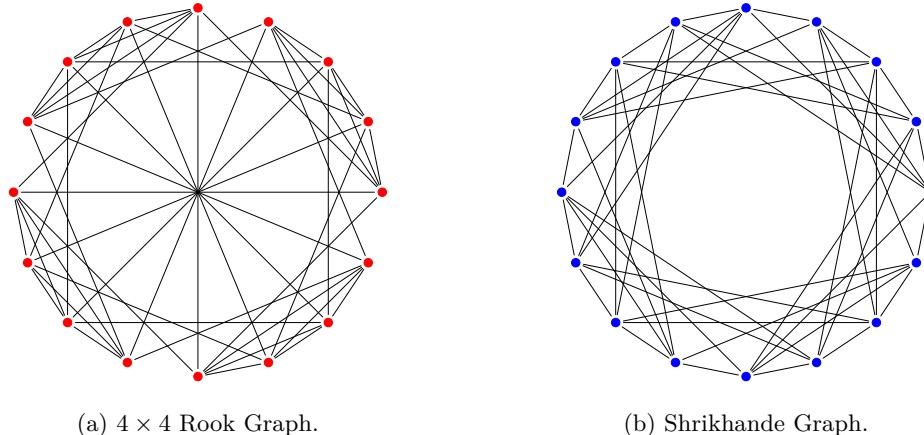

(a) $4 \times 4$ Rook Graph.      (b) Shrikhande Graph.

Figure 1: Expressive power is typically analyzed in terms of the families of non-isomorphic graphs *that models fail to distinguish*: $4 \times 4$ Rook (a) and Shrikhande (b) graphs are indistinguishable by node-only sub-graph GNNs (Frasca et al., 2022).

Another example of a challenging benchmark is the *h*-Proximity task (shown in Fig. 2), which requires the ability to capture graph properties that depend both on the graph structure (shortest path distances) and node attributes (colors) (Abboud et al., 2022). In this case, an enriched (learnable) MP-GNN with ELENE features—called ELENE-L—outperforms current baselines. In real-world benchmarks, ELENE-L matches the performance of state-of-the-art learning methods at significantly lower memory costs, as we show experimentally in §7.

The paper is organized as follows. §3 defines and motivates ELENE. §4 introduces ELENE-L. §5 describes related work and §6 analyzes expressivity. Finally, §7 evaluates our methods in four benchmarks and §8 summarizes our results.

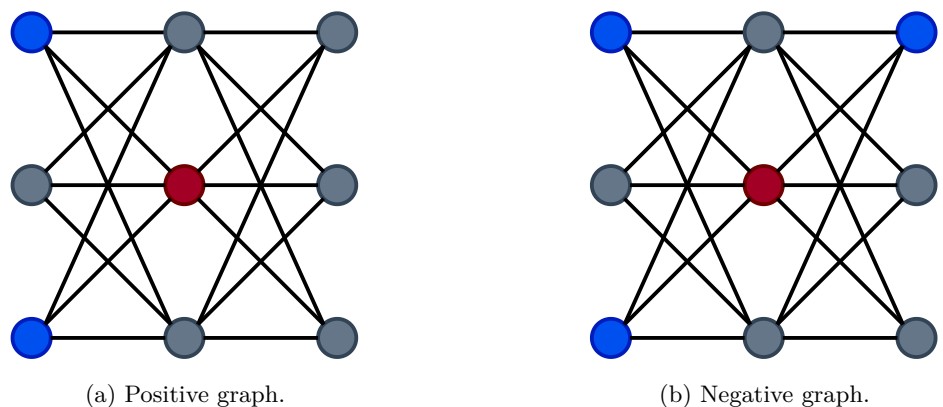

(a) Positive graph.      (b) Negative graph.

Figure 2: *h*-Proximity binary classification task—A pair of positive (a) and negative (b) 1-Proximity graph examples. An *h*-Proximity graph is positive if all red nodes have at most 2 blue neighbors up to distance *h*, and negative otherwise.

## 2 Notation and Definitions

In this work, $G = (V, E)$ denotes a graph with $n = |V|$ and $m = |E|$. $l_G(u, v)$ is the shortest path length between $u, v \in V$ in $G$. $d_G(v)$ is the degree of $v$ in $G$ and we use $d_{\max}$ for the maximum degree over all nodes in $G$. Double brackets $\{\!\{\cdot\}\!\}$ denote multi-sets while $\bigcup$ and $\bigcap$, respectively, indicate set and multi-set union and intersection. We use short-hand $x^r$ notation to signify $x$ is contained $r$ times, where $y = \{\!\{x^r\}\!\}$ reads as "$x$ appears $r$ times in $y$".

We use $\mathcal{S}_v^k = (\mathcal{V}_v^k, \mathcal{E}_v^k) \subseteq G$ for the $k$-depth induced ego-network sub-graph of $G$ centered on $v$ (abbreviated $\mathcal{S}$ in equations). We denote the maximum degree over all nodes in $\mathcal{S}_v^k$ by $d_{\max}^k$. Likewise, we use $\mathcal{S}_{\langle u,v \rangle}^k = (\mathcal{V}_u^k \bigcap \mathcal{V}_v^k, \mathcal{E}_u^k \bigcap \mathcal{E}_v^k)$ to denote the intersection of ego-networks across edge $\langle u, v \rangle$. Feature vectors are shown in **bold**, as $\mathbf{x}_v$ for node $v$, $\mathbf{x}_{\langle u,v \rangle}$ for edge $\langle u, v \rangle$, we denote vector concatenation by $\|$, and the Hadamard product by $\odot$. Finally, we represent a learnable embedding of a discrete input, e.g., degree or distance signals as $\texttt{Emb}(\cdot)$, and a learnable weight matrix as $\mathbf{W}$.

## 3 Defining ELENE

In this section, we first present the proposed edge-level encodings and then illustrate their expressive power.

### 3.1 Constructing an Edge-Level Ego-Network Encoding

The main idea behind ELENE encodings is to capture higher-order interactions that go beyond the node-centric perspective used by MP-GNNs. We look at the structure resulting not only from the ego-network of every node, but also from the combination of two ego-networks of adjacent nodes in the input graph, and design a pool of features based on that structure.

Consider the $k$-depth ($k > 1$) ego-network $\mathcal{S}_v^k$ surrounding node $v$. We may ask: *how many edges of a neighbor $u$ of $v$ reach nodes that are 1-hop closer to $v$, at the same distance as $u$, or 1-hop farther from $v$?* The proposed ELENE encodings elaborate on this idea to capture interactions between nodes and edges in ego-network sub-graphs.

More formally, consider a node $u$ contained in $\mathcal{S}_v^k$ and let $d_{\mathcal{S}}(u|v)$ count the edges from $u$ to nodes at a distance $l_{\mathcal{S}}(u, v) + p$ of $v$, $p \in \{-1, 0, +1\}$:

$$d_{\mathcal{S}}^{(p)}(u|v) = \left| (u, w) \in \mathcal{E}_v^k, \ \forall w \in \mathcal{V}_v^k : l_{\mathcal{S}}(v, w) = l_{\mathcal{S}}(u, v) + p \right|.$$

The degree of node $u$ decomposes as the sum of these *relative* degrees corresponding to these three different subsets of neighbors of $u$:

$$d_{\mathcal{S}}(u) = d_{\mathcal{S}}^{(-1)}(u|v) + d_{\mathcal{S}}^{(0)}(u|v) + d_{\mathcal{S}}^{(+1)}(u|v).$$

Fig. 3 (left) shows an example graph, with all nodes labeled with their degree and colored according to the distance to the root node of the ego network, in this case, the node in green. The plot on the right shows a degree triplet for each node, which counts the *relative* degrees, or edges closer and further to the root ($1^{\text{st}}$ and $3^{\text{rd}}$ components), together with the individual degree ($2^{\text{nd}}$ component).

Leveraging relative degrees yields ELENE, an ego-network encoding as a multi-set of quadruplets counting all instances of distance and degree triplets in sub-graph $\mathcal{S}$:

$$e_v^k = \left\{\!\!\left\{ \left( l_{\mathcal{S}}(u, v), d_{\mathcal{S}}^{(-1)}(u|v), d_{\mathcal{S}}(u), d_{\mathcal{S}}^{(+1)}(u|v) \right) \middle| \forall u \in \mathcal{V}_v^k \right\}\!\!\right\}. \tag{1}$$

We can construct an Edge (**ED**) Centric encoding analogous to the Node (**ND**) Centric encoding of Eq. 1 by also encoding edge-wise sub-graph intersections for edge $\langle u, v \rangle$ as $e_{\langle u,v \rangle}^k$ and counting quadruplets across $\mathcal{S}_{\langle u,v \rangle}^k$ with distances to both $u$ and $v$. Using **ED** or **ND** encodings leads to different expressive power, as we show formally in §6. In both cases, App. C shows that ELENE encodings are permutation invariant at the node level and equivariant at the graph level.

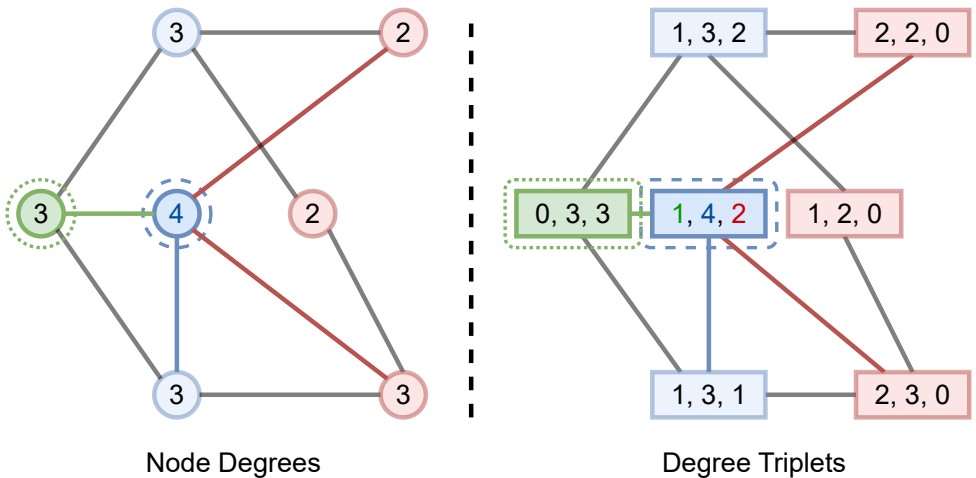

Figure 3: Example graph (right) and corresponding (left) degree triplets for nodes in the 2-hop ego-network rooted on the green node. The dashed blue node has one edge to the 0-hop root ($d_{\mathcal{S}}^{(-1)} = 1$), a degree of 4, and two edges 2-hops from the root ($d_{\mathcal{S}}^{(+1)} = 2$, red), so its degree triplet is $(1, 4, 2)$.

## 3.2 Illustrating ELENE

To illustrate ELENE, we focus on Strongly Regular graphs. An $n$-vertex graph is $d$-regular if all $n$ nodes have degree $d$, i.e., $\forall v \in V, d_G(v) = d$. An $n$-vertex $d$-regular graph is said to be Strongly Regular if there exists $\lambda, \mu \in \mathbb{N}$ such that every two adjacent nodes have $\lambda$ neighbors in common, and every two non-adjacent nodes have $\mu$ neighbors in common. We denote Strongly Regular Graphs as $\texttt{SRG}(n, d, \lambda, \mu)$. Strongly Regular graphs with equal parameters are *indistinguishable* by the 1-WL (Weisfeiler & Leman, 1968) test—a classic graph algorithm known to distinguish graphs that are not isomorphic with high probability (Babai & Kucera, 1979)—and its more powerful $k = 3$-WL variant (Arvind et al., 2020; Balcilar et al., 2021)—whose ability to distinguish graphs has been shown to be the expressivity upper-bound for node-only Sub-graph GNNs (Bevilacqua et al., 2022; Frasca et al., 2022; Zhao et al., 2022). A natural question follows: *what structural information is sufficient to distinguish SRGs?*

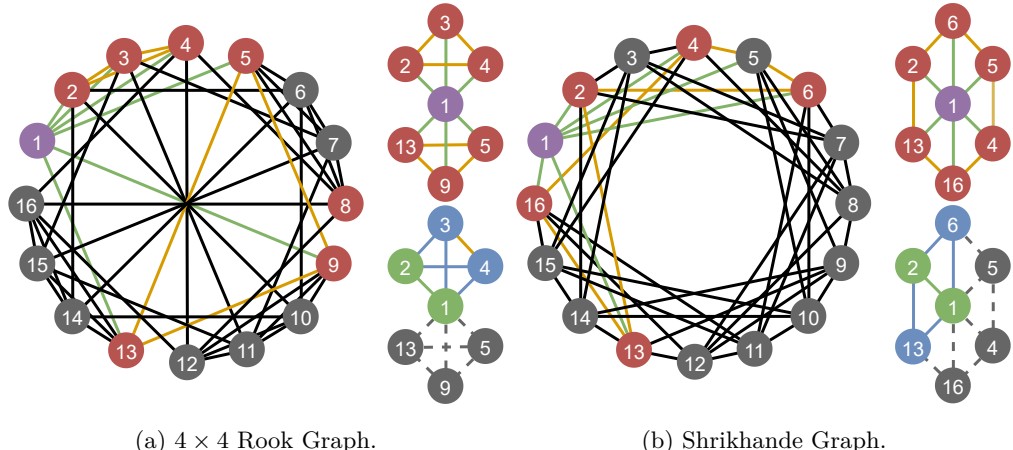

(a) $4 \times 4$ Rook Graph.        (b) Shrikhande Graph.

Figure 4: The $4 \times 4$ Rook (a) and Shrikhande (b) graphs are indistinguishable by 3-WL as SRGs with parameters $\texttt{SRG}(16, 6, 2, 2)$ (Arvind et al., 2020; Frasca et al., 2022). ELENE (**ND**, top sub-graphs) is also unable to distinguish the graphs, while ELENE (**ED**, bottom sub-graphs) counts different numbers of edges.

In Fig. 4, we show the 1-depth ego-networks $\mathcal{S}_{v_1}^{k=1}$ for the purple vertices labeled with '1' with its 1-hop neighbors colored in red (top smaller sub-graphs)[1]. We represent both graphs in terms of $n = 16$ equal sub-graphs (one per node), analyzing whether the sub-graph pairs can be distinguished. Both sub-graphs have the same number of nodes (7), edges (12), and matching degree multisets $\{\!\{3^6, 6^1\}\!\}$. Furthermore, by coloring the edges as connected to the ego-network root (in green) or connecting adjacent neighbors of the root (in orange), the count of edge colors also matches. The Node-Centric (**ND**) ELENE encoding, as shown in Eq. 1, corresponds to such a coloring and is thus unable to distinguish the pair of graphs. In §6, we formally prove this upper bound for ELENE (**ND**), which coincides with the expressive power of node-based Sub-graph GNNs.

In contrast, if we consider the 1-hop common neighbors (in blue) of adjacent nodes labeled '1' and '2', the intersecting sub-graphs are distinguishable (bottom smaller sub-graphs). Indeed, the number of edges differs between the $4 \times 4$ Rook graph (6 edges) and the Shrikhande's graph (5 edges). This corresponds to Edge-Centric ELENE (**ED**), and illustrates it has more expressive power than the Node-Centric (**ND**) one.

### 3.3 Computational Complexity

The ELENE encoding for a single node $v$ requires traversing all the edges in the ego-network $\mathcal{E}_v^k$. This can be computed via Breadth-First Search (BFS) bounded by depth $k$, which has worst-case complexity $\mathcal{O}(d_{\texttt{max}}^k)$. If $k$ is greater than the diameter in the graph, ELENE must traverse all $m$ edges for the $n$ ego-networks with each node as the root. Encoding the entire graph thus has time complexity $\mathcal{O}(n \cdot \min\{m, d_{\texttt{max}}^k\})$. Note that the more expressive edge-centric implementation requires executing the BFS from both nodes alongside an edge, with asymptotically no additional cost.

ELENE is best suited for sparse graphs, where $d_{\texttt{max}} \ll n$. For fully connected graphs, $m = |\mathcal{E}_v^k| = n \cdot (n-1)/2$ which results in time complexity $\mathcal{O}(n^3)$, matching the computational worst-case complexity of GNN-AK Zhao et al. (2022), NGNNs Zhang & Li (2021), SUN Frasca et al. (2022), ESAN Bevilacqua et al. (2022), or SPEN Mitton & Murray-Smith (2023).

In terms of memory, ELENE encodings require a sparse $3 \cdot (k+1) \cdot (d_{\texttt{max}} + 1)$-component vector for each node $v \in V$ to represent the multi-set of quadruplets in Eq. 1. Accordingly, each entry holds the count of observed relative degrees at each distance from $v$. In App. A, we describe a BFS implementation that produces a mapping of each ELENE encoding quadruplet to its frequency, and can be parallelized over $p$ processors yielding $\mathcal{O}(n \cdot \min\{m, d_{\texttt{max}}^k\}/p)$ time complexity.

## 4 Learning with ELENE: ELENE-L

We now introduce two approaches for leveraging ELENE encodings in practical learning settings—a simple concatenation of ELENE over network attributes, and a fully learnable variant called ELENE-L that updates node and edge representations during the learning process. The first approach represents ELENE multi-as sparse vectors containing frequencies of each quadruple $q$—which can be attributes concatenated to $\mathbf{x}_v$ or $\mathbf{x}_{\langle u,v \rangle}$ if processing $e_v^k$ or $e_{\langle u,v \rangle}^k$:

$$\text{ELENE}_{\texttt{vec}}^k(v)_i = \left| \left\{\!\!\left\{ q \in e_v^k \middle| f(q) = i \right\}\!\!\right\} \right|. \tag{2}$$

where $f(q)$ is an indexing function mapping each unique quadruplet to an index in the sparse vector.

The concatenation approach is the most memory efficient, using only as much memory as the encodings themselves, and can be computed once and reused during training or inference. Furthermore, this approach is directly applicable to any downstream learning model e.g., an MP-GNN, without changing its architecture.

Certain tasks, however, require structural information *within the sub-graph* to be combined with node or edge *attributes* during learning. One example are $h$-Proximity tasks, which require joint representations that integrate the ELENE encodings with attributes and structure.

---

[1]Note that for these two graphs, any node will have matching ego-networks regardless of their label—see proof for Theo. 2.

ELENE-L addresses the limitations of concatenating ELENE with node and edge attributes by learning over both *structures* and *attributes* at once. ELENE-L learns non-linear functions ($\Phi$, e.g. a Dense Neural Network—DNN) to represent nodes, edges, and Node or Edge-Centric sub-graphs by representing $u$ in $\mathcal{S}_v^k$ via a learnable function $\Phi_{\texttt{nd}}$[2]:

$$\Phi_{\texttt{nd}}^t(u|v) = \Phi_{\texttt{nd}}\left(\mathbf{x}_v^t \middle\| \mathbf{x}_u^t \middle\| \texttt{Emb}(u|v)\right), \tag{3}$$

where $\mathbf{x}_v^t$ and $\mathbf{x}_u^t$ are features of $u$ and $v$ at time-step $t$ (i.e. after $t$ layers, such that $t=0$ are 'input' features), and $\texttt{Emb}(u|v)$ is a learnable embedding of ELENE encodings which we describe in §4.1. As with ELENE, we produce a learnable representation of an edge $\langle u, w\rangle$ in $\mathcal{S}_v^k$ via a learnable $\Phi_{\texttt{ed}}$:

$$\Phi_{\texttt{ed}}^t(u, w|v) = \Phi_{\texttt{ed}}\left(\mathbf{x}_v^t \middle\| \mathbf{x}_{\langle u, w\rangle}^t \middle\| \mathbf{x}_u^t \odot \mathbf{x}_w^t \middle\| \texttt{Emb}(u, w|v)\right).$$

The representation of the Node-Centric ego-network root at time $t$ is a learnable $\Phi_{\texttt{ND}}$ applied over the aggregation of every node and edge in the sub-graph given a pooling function ($\sum$):

$$\Phi_{\texttt{ND}}^t(v) = \Phi_{\texttt{ND}}\left(\mathbf{x}_v^t \middle\| \sum_u^{\mathcal{V}_v^k} \Phi_{\texttt{nd}}^t(u|v) \middle\| \sum_{\langle u, w\rangle}^{\mathcal{E}_v^k} \Phi_{\texttt{ed}}^t(u, w|v)\right). \tag{4}$$

Similarly, the Edge-Centric representation for edge $\langle u, v\rangle$ at time $t$ is a learnable $\Phi_{\texttt{ED}}$ consuming the aggregation over ego-networks containing the edge, as shown in Fig. 5:

$$\Phi_{\texttt{ED}}^t(u, v) = \Phi_{\texttt{ED}}\left(\sum_w^{\mathcal{V}_{\langle u, v\rangle}^k} \Phi_{\texttt{ed}}^t(u, v|w)\right). \tag{5}$$

Node and edge representations at $t+1$ update via a learnable parameter $\gamma$ gating the flow of ELENE updates:

$$\mathbf{x}_v^{t+1} = \mathbf{x}_v^t + \gamma_{\texttt{ND}} \cdot \Phi_{\texttt{ND}}^t(v). \tag{6}$$

We follow the same update-rule at the edge level:

$$\mathbf{x}_{\langle u, w\rangle}^{t+1} = \mathbf{x}_{\langle u, w\rangle}^t + \gamma_{\texttt{ED}} \cdot \Phi_{\texttt{ED}}^t(u, w). \tag{7}$$

We may use $\mathbf{x}_v^{t+1}$ and $\mathbf{x}_{\langle u, w\rangle}^{t+1}$ directly in the downstream task, or as inputs into an MP-GNN layer during learning—boosting its expressivity. We follow the latter approach in this work.

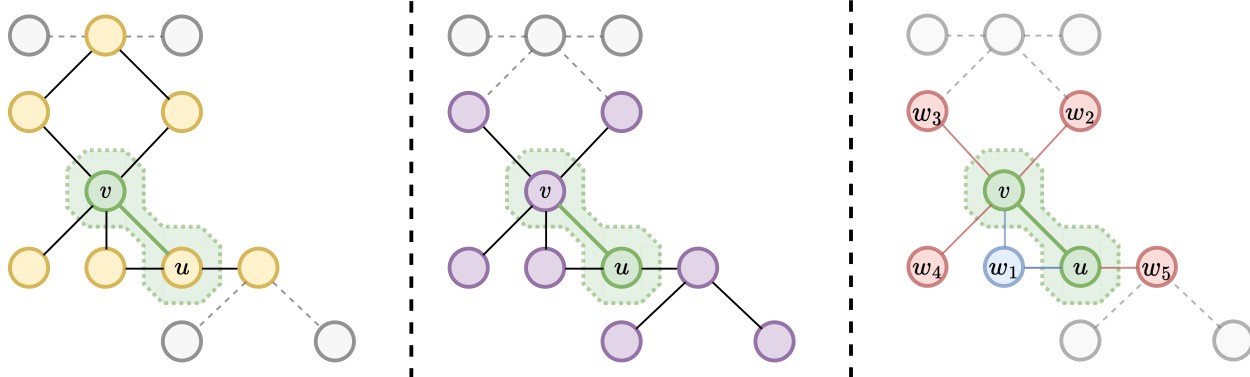

Figure 5: $k$-depth ego-network intersection following Eq. 5 for the green edge. The ego-networks of $u$ and $v$ (yellow (left) and purple (center) respectively), intersect on five nodes around $\langle u, v\rangle$ (dotted, right). We show $\mathcal{V}_{\langle u, v\rangle}^{k=2} = \{u, v, w_1, w_2, w_3, w_4, w_5\}$ (right), indicating nodes reachable in 0 or 1-hops, exactly 1-hop or 1 or 2-hops from $u$ and $v$.

---

[2]We compress notation by using $\Phi^t$ for the output of $\Phi$ at step $t$.

### 4.1 Defining ELENE-L Embeddings

The representations in Eq. 6 and Eq. 7 leverage attributes and ELENE encodings through $\texttt{Emb}(u|v)$ and $\texttt{Emb}(u,w|v)$. To define ELENE-L embeddings, three hyper-parameters determine the shapes of embedding matrices: $\omega$, the length of the embedding vectors; $\rho$, the max. degree to be encoded (by default, $\rho = d_{\texttt{max}}$); and $k$, the maximum distance to be encoded (i.e., the ego-network depth). For the quadruplet of $u$, we abbreviate:

$$q_u = (l_u, d_u^1, d_u^2, d_u^3) = \left(l_{\mathcal{S}}(u,v), d_{\mathcal{S}}^{(-1)}(u|v), d_{\mathcal{S}}(u), d_{\mathcal{S}}^{(+1)}(u|v)\right)$$

as defined in Eq. 1 and *jointly* embed distance and relative degrees. The embedding $\texttt{Emb}(u|v)$ of $u$ in $\mathcal{S}_v^k$ is given by:

$$\texttt{Emb}\left[l_u, d_u^1, d_u^2, d_u^3\right] = \left(\mathbf{W}_{(l_u,d_u^1)}^{1,\texttt{nd}} \middle\| \mathbf{W}_{(l_u,d_u^2)}^{2,\texttt{nd}} \middle\| \mathbf{W}_{(l_u,d_u^3)}^{3,\texttt{nd}}\right), \tag{8}$$

where $\mathbf{W}^{1,\texttt{nd}}$, $\mathbf{W}^{2,\texttt{nd}}$, and $\mathbf{W}^{3,\texttt{nd}} \in \mathbb{R}^{S \times \omega}$ are three node embedding matrices with $S = (\rho + 1) \cdot (k + 1)$ entries—one for each distance and relative degree pair. A visual representation of the attributes and ELENE encodings of $u$ is shown in Fig. 6.

To embed edge $\langle u, w\rangle$ in $\mathcal{S}_v^k$, we use the quadruplets of $u$ and $w$, $q_u$ and $q_w$, and increase the granularity of distances to capture the *relative* direction of the edge, following Fig. 4:

$$\delta_{uw} = l_u - l_w + 1 \in \{0, 1, 2\}.$$

We embed $\langle u, w\rangle$ in a permutation-invariant manner, summing embeddings bidirectionally so $\texttt{Emb}(u,w|v) = \texttt{Emb}(w,u|v)$:

$$\texttt{Emb}(u,w|v) = \left(\mathbf{W}_{(l_u,\delta_{uw},d_u^1)}^{1,\texttt{ed}} \middle\| \mathbf{W}_{(l_u,\delta_{uw},d_u^2)}^{2,\texttt{ed}} \middle\| \mathbf{W}_{(l_u,\delta_{uw},d_u^3)}^{3,\texttt{ed}}\right) + \left(\mathbf{W}_{(l_w,\delta_{wu},d_w^1)}^{1,\texttt{ed}} \middle\| \mathbf{W}_{(l_w,\delta_{wu},d_w^2)}^{2,\texttt{ed}} \middle\| \mathbf{W}_{(l_w,\delta_{wu},d_w^3)}^{3,\texttt{ed}}\right). \tag{9}$$

$\mathbf{W}^{1,\texttt{ed}}$, $\mathbf{W}^{2,\texttt{ed}}$, and $\mathbf{W}^{3,\texttt{ed}} \in \mathbb{R}^{3 \times S \times \omega}$ are edge-level embedding matrices with $3\times$ more entries to represent the three possible values of $\delta_{uw}$.

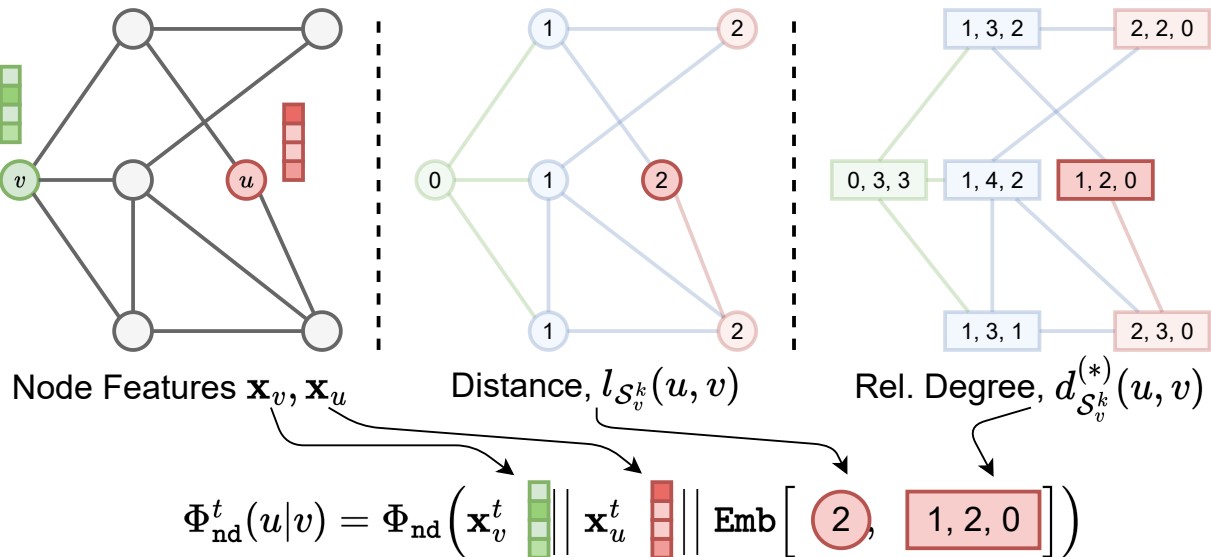

Figure 6: ELENE-L encoding of $u$ in the $k = 2$ ego-network of $v$. The representation contains the feature vectors of both nodes ($\mathbf{x}_v$ and $\mathbf{x}_u$, left), the distance information of $u$ to $v$ ($2$, center) and the relative degree information ($[1, 2, 0]$, right).

## 5  Related Work

In this section, we connect related work with ELENE encodings and the practical applications of ELENE and ELENE-L in §4. Per §1, the expressivity of MP-GNNs is often studied through the 1-WL test and its more powerful $k$-WL variants. Despite great successes in many domains (Duvenaud et al., 2015; Battaglia et al., 2016; Gilmer et al., 2017; Ying et al., 2018), the GIN architecture (Xu et al., 2019) showed that one-hop MP-GNNs are at most as expressive as 1-WL. This increased interest in expressive power within the community—in the formal study of MP-GNNs (Papp & Wattenhofer, 2022), and to boost message-passing with spectral (Balcilar et al., 2021), positional (You et al., 2019; Li et al., 2020; Abboud et al., 2022), path-level (Eliasof et al., 2022; Michel et al., 2023), sub-graph (Nikolentzos et al., 2020; Zhang & Li, 2021; Bevilacqua et al., 2022; Frasca et al., 2022; Mitton & Murray-Smith, 2023), structural signals (Morris et al., 2019; Bodnar et al., 2021), or their combination (Ying et al., 2021; Zhao et al., 2022; Dwivedi et al., 2022; Rampášek et al., 2022). We now discuss the theoretical ability of models to express certain computations—*expressivity* in the abstract—and empirical performance and architectures of graph learning methods.

— **Expressivity.** The most common framework to study expressivity are the $k$-WL tests and its variants (Morris et al., 2019). Recent research has also focused on other perspectives, such as matrix languages (Balcilar et al., 2021), or the GD-WL test (Zhang et al., 2023)—which reframes expressivity in terms of graph biconnectivity, capturing the ability to identify cut nodes and edges. Shortest Path Neural Networks (SPNNs) (Abboud et al., 2022) introduced a model aggregating across shortest-path distances, but not edges or messages across neighbors, whose expressivity differs from 1-WL and addresses the over-squashing problem (Alon & Yahav, 2021). Finally, another approach has been through 2-variable counting logics (Barceló et al., 2020; Grohe, 2021)—studying what Boolean statements MP-GNNs can express.

ELENE builds on previous expressivity analyses by presenting features that can distinguish challenging 3-WL equivalent graphs—`SRG`s. In §6, we will show that ELENE can fully identify between `SRG`s with different parameters, and prove that an ELENE-L model can emulate SPNNs. Furthermore, in §7 we empirically evaluate our models on the $h$-Proximity tasks and explore whether *structural ($k$-WL) expressivity is all we need*. We find ELENE-L outperforms previous strong baselines of SPNNs and Graphormers (Ying et al., 2021; Abboud et al., 2022) while simply concatenating ELENE encodings underperforms, showing that *expressivity* without *attributes* is insufficient for certain tasks.

— **Boosting Graph Neural Models.** Besides studying flavors of expressivity, researchers have also focused on improving performance for MP-GNNs and Graph Transformers. We summarize the most relevant families of novel network architectures in connection with ELENE and ELENE-L:

— **Sub-graph MP-GNNs.** ELENE is most related to equivariant, sub-graph methods—including $k$-hop GNNs (Nikolentzos et al., 2020), Structural MP-GNNs (**SMP**) (Vignac et al., 2020), NestedGNNs (**NGNNs**) (Zhang & Li, 2021), Identity-GNNs (**ID-GNN**) (You et al., 2021), Equivariant Subgraph Aggregation Networks (**ESAN**) (Bevilacqua et al., 2022), Ordered Subgraph Aggregation Networks (**OSAN**) (Qian et al., 2022), GNN-As-Kernel (**GNN-AK**) (Zhao et al., 2022), Shortest Path Neural Networks (**SPNN**) (Abboud et al., 2022), Subgraph Union Networks (**SUN**) (Frasca et al., 2022), and Subgraph Permutation Equivariant Networks (**SPEN**) (Mitton & Murray-Smith, 2023). By encoding structural attributes of the ego-network sub-graph, ELENE captures similar signals as GNN-AK's centroid encodings. However, ELENE-L extends node and edge representations within sub-graphs *first*, and then feeds sub-graph aware data into a GNN—rather than applying a GNN on the sub-graph and aggregating its outputs as in NGNNs and GNN-AK. During learning, ELENE-L resembles SPEN and ESAN with the `EGO+` policy with node marking, as the root of the ego-network is implicitly marked by the relative degree and distance pairs.

These sub-graph GNNs involve processing the sub-graphs during training and inference, which is avoided by approaches like IGEL (Alvarez-Gonzalez et al., 2022), GSNs (Bouritsas et al., 2023) or ESC-GNN (Yan et al., 2023), and also ELENE—as they add sub-structure information without executing a GNN in the sub-graph. Theo. 1 shows ELENE encodings are a superset of sparse IGEL vectors. ELENE requires no choice of substructure to count. In contrast, GSNs require counting $k$-node structures which has an exponential cost in $k$. Finally, ESC-GNNs also use structural degree and distance signals directly as inputs. However, ELENE-L learns additional embeddings from the structural encodings rather than using them as static features.

Other approaches instead tackle the representation task by learning to select sub-graphs, such as MAG-GNN (Kong et al., 2023) or Policy-Learn (Bevilacqua et al., 2023)—for which ELENE signals could act as additional features. Finally, ELENE-L can be understood as a graph rewiring approach as recently exemplified by Dynamic Graph Rewiring (**DRew**) (Gutteridge et al., 2023), since each ELENE-L layer can be independently parameterized to connect nodes via ego-networks *and* edge-level sub-graphs—adding virtual edges between vertices that are $k$-hops away, and also passing signals across adjacent nodes (i.e. edges) whose $k$-depth ego-networks intersect. In this work, we only explore the impact of *static* edge-level rewiring through relative degrees and Node or Edge-Centric sub-graphs.

The key difference with the aforementioned methods is that ELENE-L captures edge-level information both in the encoding and during learning, as per Fig. 4. §6.2, shows edge-level signals boost expressivity and corroborates results from **SUN** that node-only sub-graph models are upper-bounded by the 3-WL test (Frasca et al., 2022). In §7.2, we experimentally validate that ELENE-L (**ED**) but not (**ND**) reaches 100% accuracy on SR25, a challenging SRG dataset only solved before without graph perturbations by $\mathcal{O}(n^2)$ **PPGN-AK** (Maron et al., 2019; Zhao et al., 2022), and partially by **SPEN** (Mitton & Murray-Smith, 2023), which distinguished 97% of non-isomorphic pairs.

— **Perturbation methods.** Beyond sub-graph methods, random perturbations of the graph structure like DropGNN (Papp et al., 2021), Random Node Initializations (Abboud et al., 2021), or paths from random walks (Eliasof et al., 2022) have also shown surprising performance in expressivity tasks. Furthermore, random-walks based methods have been shown to be effective at capturing structural information, including positional information (**RWPE**) (Dwivedi et al., 2022). Although ELENE in its current definition does not consider graph perturbations or stochastic features, the underlying quadruplets can be easily adapted to ignore dropped-out nodes or edges, and can be seamlessly combined with random node initializations or global positional encodings.

— **Graph Transformers.** Similarly, the extension of Transformer models to graph tasks has led to increased research interest, notably with the introduction of Graphormer (Ying et al., 2021)—which included positional and degree encoding similar to ELENE, but using only *absolute* in/out degrees. More recently, Pure Graph Transformers (Kim et al., 2022) removed graph-specific architecture choices, directly encoding nodes and edges as tokens processed through self-attention and a global read-out.

Finally, a series of works have yielded high-performance recipes for graph transformers such as GPS (Rampášek et al., 2022)—combining strong inductive biases from MP-GNNs, as well as global and local encoding to build high-performance Graph Transformers. Transformers on graphs can be understood as *fully-connected* graph processors, and it has been shown that Graphormers can be emulated through an SPNN (Abboud et al., 2022). In §6.3, we show that ELENE-L can, in turn, emulate an SPNN—and transitively a Graphormer. We consider the analysis of edge-level ELENE signals in Graph Transformers as future work, focusing our study on MP-GNN architectures.

## 6 Expressive Power

We now analyze the expressive power of ELENE—formally answering our question on *which information is sufficient to distinguish SRGs.* We extend recent results on IGEL, a sparse vector encoding similar to ELENE (Alvarez-Gonzalez et al., 2022) and show that Edge-Centric and Node-Centric ELENE are strictly more expressive than previous methods relying on degrees and distances by comparing their expressivity on SRGs. We then show that ELENE-L is at least as expressive as ELENE, and prove that ELENE-L (**ED**) is more expressive than ELENE-L (**ND**) and ELENE (**ND**). Finally, we connect our framework with SPNNs (Abboud et al., 2022), showing that the latter can be expressed by an instance of node-centric ELENE-L without edge-degree information—motivating our analysis on *attributed* tasks in §7.

### 6.1 Expressive Power of ELENE

Previous work has shown that encoding-based and sub-graph MP-GNN methods are limited in their ability to distinguish 3-WL equivalent SRGs (Arvind et al., 2020; Balcilar et al., 2021; Alvarez-Gonzalez et al., 2022; Frasca et al., 2022). Recently, Alvarez-Gonzalez et al. (2022) presented IGEL—a simple, sparse node feature

vector containing counts of distance and degree tuples in an ego-network, showing it is strictly more expressive than the 1-WL test. Following Eq. 2, ELENE multi-sets may also be represented as sparse vectors—which can then be used as feature vectors, but also to distinguish ego-network sub-graphs.

We build on top of the results from (Alvarez-Gonzalez et al., 2022) and show ELENE is at least as expressive as IGEL. We then find an upper-bound of expressivity for IGEL, which is at most able to distinguish between $n$, $d$ or $\lambda$ parameters of SRGs, but not $\mu$, and show Node-Centric and Edge-Centric ELENE is strictly more expressive than IGEL on SRGs as it can explicitly encode all SRG parameters by counting edges:

**Theorem 1.** *Node-Centric ELENE is at least as expressive as IGEL (Alvarez-Gonzalez et al., 2022), and transitively more expressive than 1-WL.*

*Proof.* The IGEL encoding in Alvarez-Gonzalez et al. (2022) is a simpler version of Eq. 2 that only considers distance ($l_u$) and *absolute* degree ($d_u^2$):

$$\text{IGEL}_{\texttt{vec}}^k(v)_i = \left| \left\{\!\!\left\{ (l_u, d_u^1, d_u^2, d_u^3) \in e_v^k \Big| f'(l_u, d_u^2) = i \right\}\!\!\right\} \right|.$$

where $f'(l_u, d_u^2)$ is a bijective function that does not consider *relative* degrees, in contrast with ELENE's $f$. Thus, for any ego-network, ELENE includes all information required to construct IGEL vectors, so it is at least as expressive as IGEL. □

**Theorem 2.** *ELENE (**ND**) encodes and distinguishes SRGs with different parameters of $n$, $d$, $\lambda$ and $\mu$.*

*Proof.* Consider $\text{SRG}(n, d, \lambda, \mu) = (V, E)$. The maximum diameter of an SRG is 2 (Brouwer & Van Maldeghem, 2022), so we focus on the case where $k = 2$. The ELENE (**ND**) encoding of $v \in V$ according to Eq. 1 is:

$$e_v^2 = \left\{\!\!\left\{ \Big(0, 0, d, d\Big)^1, \Big(1, 1, d, d\text{-}\lambda\text{-}1\Big)^d, \Big(2, d-\mu, d, 0\Big)^{n\text{-}d\text{-}1} \right\}\!\!\right\}$$

By definition, any Node-Centric ego-network in an SRG has a single root with $d$ neighbors, $d$ neighbors with one edge with the root and $d - \lambda - 1$ edges to the next layer, and the remaining $n - d - 1$ non-adjacent nodes to the root each have $d - \mu$ edges with the $d$ neighbors of the root. Consider $\text{SRG}'(n', d', \lambda', \mu') = (V', E')$. If any of the parameters between SRG and SRG′ differ, so will $e_v^2$ from $e_{v'}^2$. This is not the case for IGEL, which can at most capture $n$, $d$, and $\lambda$:

$$\text{IGEL}_v^1 = \left\{\!\!\left\{ \Big(0, d\Big)^1, \Big(1, 1 + \lambda\Big)^d \right\}\!\!\right\}$$

$$\text{IGEL}_v^2 = \left\{\!\!\left\{ \Big(0, d\Big)^1, \Big(1, d\Big)^d, \Big(2, d\Big)^{n\text{-}d\text{-}1} \right\}\!\!\right\}$$

Thus, ELENE (**ND**) can encode and distinguish all parameters of SRGs—outperforming IGEL. However, ELENE (**ND**) *cannot distinguish* non-isomorphic SRGs when $n = n'$, $d = d'$, $\lambda = \lambda'$, and $\mu = \mu'$. □

**Corollary 1.** *ELENE (**ND**) is more expressive than IGEL and 1-WL, per Theo. 1 & Theo. 2.*

**Corollary 2.** *ELENE (**ND**) signals at the node-level are not capable of distinguishing between non-isomorphic SRGs with equal parameters—e.g. the graphs in Fig. 4.*

**Proposition 1.** *ELENE (**ED**, leveraging both $e_v^k \ \forall v \in V$ and $e_{\langle u,v \rangle}^k \ \forall (u, v) \in E$) is strictly more expressive than ELENE (**ND**), as it can distinguish the pair of graphs in Fig. 4.*

## 6.2 Expressive Power of ELENE-L

**Theorem 3.** *ELENE-L with the sum as the pooling operator is at least as expressive as ELENE.*

*Proof.* We first show ELENE-L (**ND**) is at least as expressive as ELENE (**ND**). We then show that the **ED** variants are at least as expressive as **ND** variants (Prop. 2), and show through Fig. 4 that ELENE-L (**ED**) is more powerful than Node-Centric ELENE-L (**ND**).

**On** ELENE-L (**ND**). $\forall v \in V$, the ELENE-L$(\mathbf{x}_v^t)$ representation of $v$ is given by $\Phi_{\mathtt{ND}}^t(v)$ as per Eq. 4. $\Phi_{\mathtt{ND}}^t(v)$ is the result of applying $\Phi_{\mathtt{ND}}$ to the concatenation of $\mathbf{x}_v^t$ and the combined representations of every $u \in \mathcal{V}_v^k$ and $\langle u, w \rangle \in \mathcal{E}_v^k$. Let $\Phi_{\mathtt{out}}$ and $\Phi_{\mathtt{nd}}$ be the identity function, we exclude edge-level information by discarding the output of $\Phi_{\mathtt{ed}}$. We now expand $\hat{\Phi}_{\mathtt{ND}}^t(v)$, which is $\Phi_{\mathtt{ND}}^t(v)$ with the changes to the learnable $\Phi$:

$$\hat{\Phi}_{\mathtt{ND}}^t(v) = \left( \mathbf{x}_v^t \middle\| \sum_u^{\mathcal{V}_v^k} \left( \mathbf{x}_v^t \middle\| \mathbf{x}_u^t \middle\| \mathtt{Emb}(u|v) \right) \right).$$

We discard repeated $\mathbf{x}_v^t$ terms, and rewrite the representation of $v$, distributing the sum over the concatenated vector:

$$\hat{\Phi}_{\mathtt{ND}}^t(v) = \left( \mathbf{x}_v^t \middle\| \sum_u^{\mathcal{V}_v^k} \mathbf{x}_u^t \middle\| \sum_u^{\mathcal{V}_v^k} \mathtt{Emb}(u|v) \right).$$

Let $\mathbf{W}^{1,\mathtt{nd}}, \mathbf{W}^{2,\mathtt{nd}}, \mathbf{W}^{3,\mathtt{nd}} \in \mathbb{R}^{S \times S}$ used by $\mathtt{Emb}$ be identity matrices so every relative degree and distance pair out of $S = d_{\mathtt{max}} \cdot (k+1)$ has a single position in $\mathbf{W}$. By using the sum as the pooling function, we obtain the frequency of each relative degree and distance pair, matching ELENE in Eq. 2. Thus, $\hat{\Phi}_{\mathtt{out}}^t(v)$ contains the information contained in the ELENE multi-set, reaching at least the same expressivity. $\qquad \square$

**Proposition 2.** *ELENE-L (**ED**) variants with the sum as the pooling operator are at least as expressive as ELENE.*

**On** ELENE-L (**ED**). We had discarded $\Phi_{\mathtt{ed}}$, showing **ED** variants are at least as expressive as **ND** variants, since the concatenation of edge-level information can only match or boost expressivity. Thus, (**ND**) and (**ED**) variants of ELENE-L are as expressive as ELENE. $\qquad \square$

**Theorem 4.** *ELENE-L (**ED**) is more expressive than ELENE-L (**ND**) and ELENE (**ND**).*

*Proof.* There is at least a pair of non-isomorphic SRGs that ELENE-L (**ED**) can distinguish. In §3.2, we show that the Shrikhande and $4 \times 4$ Rook graphs (Arvind et al., 2020; Balcilar et al., 2021) (parametrized as $\mathtt{SRG}(16, 6, 2, 2)$) can be distinguished by Edge-Centric counts that ELENE-L (**ED**) captures despite being undistinguishable by 3-WL (by implementing Eq. 1 at the *edge* level). Following from Theo. 1 and Theo. 2, both graphs are indistinguishable by ELENE (**ND**) or ELENE-L (**ND**), as well as sub-graph GNNs like GNN-AK or SUN.

Intuitively, SRGs are indistinguishable with node-centric $k$ ego-network sub-graph encodings when $k \in \{1, 2\}$ since all nodes produce identical representations, as shown in Theo. 2. However, the graphs can be distinguished by edge-level information as per Eq. 7, as the intersection of $k$-depth ego-networks for $\langle v_1, v_2 \rangle$ differ in edge counts between both SRGs—as observed in §3.

We can see the $4 \times 4$ Rook Graph has 6 edges (i.e. $|\mathcal{E}_{\langle v_1, v_2 \rangle}^k)| = 6$) while the Shrikhande graph has $|\mathcal{E}_{\langle v_1, v_2 \rangle}^k)| = 5$, hence the graphs are distinguishable by ELENE-L (**ED**), but not ELENE-L (**ND**) or ELENE (**ND**). $\qquad \square$

## 6.3 Linking ELENE and Shortest Path Neural Networks.

**Remark 1.** *A Graphormer with max. shortest path length $M$ and global readout is an instance Shortest Path Neural Networks (SPNNs) with $k = M - 1$ depth (Abboud et al., 2022).*

**Theorem 5.** *ELENE-L (**ND**) is as expressive as Shortest Path Neural Networks (SPNNs), and transitively, Graphormers.*

*Proof.* Let $\mathcal{L}_G^k(v) = \{u | u \in V \wedge l_G(u, v) = k\}$ be the nodes in $G$ exactly at distance $k$ of $v$. In Abboud et al. (2022), a $k$-depth SPNN updates the hidden state of node $v$ an aggregation over the $1, ..., k$ exact-distance neighbourhoods:

$$\mathbf{x}_v^{t+1} = \Phi_{\mathtt{sp}} \left( (1 + \epsilon) \cdot \mathbf{x}_v^t + \sum_{i=1}^k \alpha_i \sum_{u \in \mathcal{L}_G^i(v)} \mathbf{x}_u^t \right). \tag{10}$$

We show that ELENE-L (**ND**) can implement SPNNs. First, let $\gamma_{\mathtt{nd}} = 1$ in Eq. 6, such that:

$$\mathbf{x}_v^{t+1} = \Phi_{\mathtt{ND}}^t(v) = \Phi_{\mathtt{ND}}\left(\mathbf{x}_v^t \,\Big\|\, \sum_u^{\mathcal{V}_v^k} \Phi_{\mathtt{nd}}^t(u|v) \,\Big\|\, \sum_{\langle u,w \rangle}^{\mathcal{E}_v^k} \Phi_{\mathtt{ed}}^t(u,w|v)\right).$$

We drop $\Phi_{\mathtt{ed}}^t(u,w|v)$ as SPNNs ignore edge-level signals[3]. Let $\Phi_{\mathtt{ND}}(\cdot)$ be composed of two functions $\Phi_{\mathtt{sp}}(g(\cdot))$[4] where:

$$g\left(\mathbf{x}_v^t \,\Big\|\, \sum_u^{\mathcal{V}_v^k} \Phi_{\mathtt{nd}}^t(u|v)\right) = \left((1+\epsilon) \cdot \mathbf{x}_v^t + \sum_u^{\mathcal{V}_v^k} \Phi_{\mathtt{nd}}^t(u|v)\right).$$

We replace $\Phi_{\mathtt{ND}}$ by $\Phi_{\mathtt{sp}}$ and $g(\cdot)$, and expand $\Phi_{\mathtt{nd}}^t(u|v)$:

$$\mathbf{x}_v^{t+1} = \Phi_{\mathtt{sp}}\left((1+\epsilon) \cdot \mathbf{x}_v^t + \sum_u^{\mathcal{V}_v^k} \Phi_{\mathtt{nd}}\left(\mathbf{x}_v^t \,\Big\|\, \mathbf{x}_u^t \,\Big\|\, \mathtt{Emb}(u|v)\right)\right).$$

We then instantiate $\Phi_{\mathtt{nd}}(\cdot)$ as:

$$\Phi_{\mathtt{nd}}(\cdot) = \sum_i^k \alpha_i \cdot \mathtt{if}[i = l_{\mathcal{S}}(u,v)] \cdot \mathbf{x}_u^t$$

$\mathtt{if}[\cdot]$ can be implemented through the distance and degree signals in $\mathtt{Emb}$, such we can check if the node distance matches a specific value[5]. Substituting in $\mathbf{x}_v^{t+1}$ above yields:

$$\mathbf{x}_v^{t+1} = \Phi_{\mathtt{sp}}\left((1+\epsilon) \cdot \mathbf{x}_v^t + \sum_u^{\mathcal{V}_v^k} \sum_i^k \alpha_i \cdot \mathtt{if}[i = l_{\mathcal{S}}(u,v)] \cdot \mathbf{x}_u^t\right),$$

which is equivalent to Eq. 10, and shows ELENE-L (**ND**) can learn like SPNNs—and, transitively through Rem. 1, that Graphormers can be emulated by ELENE-L (**ND**). □

# 7 Experimental Results

We now study the effect of introducing ELENE and ELENE-L in a variety of graph-level settings, evaluating where *purely structural* ELENE encodings underperform ELENE-L, and the practical impact of ELENE variants in terms of model performance, training time, and memory costs. We describe our experimental protocol in §7.1 and provide reproducible code, hyper-parameters, and analysis scripts through Github[6] for four experimental benchmarks:

**A) Expressivity**. Evaluates whether models distinguish non-isomorphic graphs (on 1-WL EXP (Abboud et al., 2021) and 3-WL SR25 (Balcilar et al., 2021) equiv. datasets), count sub-graphs (in RandomGraph (Chen et al., 2020)), and evaluate graph-level properties (Corso et al., 2020).

**B) Proximity**. Measures whether models learn long-distance *attributed* node relationships in $h$-Proximity datasets (Abboud et al., 2022).

**C) Real World Graphs**. Evaluates performance on five large-scale graph classification/regression datasets from Benchmarking GNNs (ZINC, CIFAR10, PATTERN) (Dwivedi et al., 2020), and the Open Graph Benchmark (MolHIV, MolPCBA) (Hu et al., 2020a).

**D) Memory Scalability**. Evaluates the memory consumption of ELENE-L on $d$-regular graphs, varying $n$ and $d_{\mathtt{max}}$ to validate the algorithmic complexity analysis in §3.3 and comparing with the memory consumption of GIN-AK, GIN-AK$^+$ and SPEN (Mitton & Murray-Smith, 2023).

---

[3]Including edge-level signals may bring ELENE-L (**ED**) to parity with Pure Graph Transformers (Kim et al., 2022). We do not explore this connection.

[4]$g(\cdot)$ is a linear combination over concatenated input vectors, learnable by a first layer of $\Phi_{\mathtt{out}}$ without activations.

[5]This is not necessary during learning: a one-hot 'decoder' can be implemented using a two-layer perceptron with ReLU activations.

[6]https://github.com/nur-ag/ELENE

## 7.1 Experimental Protocol

**Reporting.** When reported in the original studies, we show stddevs for experiments with more than two runs following (Zhao et al., 2022), and highlight best-performing models per task in **underlined bold**. ELENE denotes Eq. 2 as additional features, while ELENE-L denotes the representations of Eq. 6 and Eq. 7. **(ED)** denotes ELENE-L with Edge-Centric signals, while **(ND)** denotes a Node-Centric variant that ignores edge information for ablation studies. '$^\dagger$' indicates results from the literature.

**Environment.** Experiments ran on a shared server with a 48GB Quadro RTX 8000 GPU, 40 CPU cores and 502GB RAM. Each individual job has a limit of 96GB RAM and 8 CPU cores. To measure memory and time costs without sharing resources, we also reproduced our experiments on real-world graphs on a SLURM cluster with nodes equipped with 22GB Quadro GPUs. Finally, scalability experiments ran on Tesla T4 GPUs with 15.11GB of VRAM to validate our approach on consumer hardware.

**Experimental Setup.** We explore sub-sets of ELENE hyper-parameters via grid search with $k \in \{0, 1, 2, 3, 5\}$ parameter ranges for ELENE and ELENE-L, and test the ED/ND variants for ELENE-L with embedding params. $\omega \in \{16, 32, 64\}$, $\rho = d_{\texttt{max}}$, using masked-mean pooling for stability. For $h$-Proximity (Abboud et al., 2022), we compare against SPNNs (Abboud et al., 2021) and Graphormer(Ying et al., 2021) as originally reported. For Expressivity and Real World Graphs, we reuse hyper-parameters and splits from GIN-AK$^+$ in Zhao et al. (2022) without architecture search, comparing against strong MP-GNN baselines from literature where GNN-AK$^+$ underperforms: CIN (Bodnar et al., 2021) for ZINC and SUN (Frasca et al., 2022) for sub-graph counting. We choose GINE (Hu et al., 2020b), an edge-aware variant of GIN (Xu et al., 2019), as our base MP-GNN given that GIN-AK$^+$ outperforms its uplifted counterparts for GCN-AK$^+$ and PNA-AK$^+$ (Kipf & Welling, 2017; Corso et al., 2020; Zhao et al., 2022), without running into out-of-memory issues like PPGN (Maron et al., 2019) in the PPGN-AK instantiation. Finally, for scalability we compare with GNN-AK on benchmark datasets Zhao et al. (2022) and SPEN Mitton & Murray-Smith (2023).

**Experimental Objectives.** We connect *expressivity* and its relation to graph *attributes*, comparing against methods that *do not* perturb graph structure, e.g. DropGNN (Papp et al., 2021); leverage random walks, e.g. RWPE (Dwivedi et al., 2022); or require costly pre-processing e.g. $\mathcal{O}(n^3)$ spectral eigendecompositions, such as GNNML3, LWPE, or GraphGPS (Balcilar et al., 2021; Dwivedi et al., 2022; Rampášek et al., 2022). Per §6.3, ELENE relates to Graphormers via SPNNs, so we focus on sub-graph GNNs and SPNNs.

## 7.2 Expressivity

We test ELENE on four MP-GNN expressivity datasets, with results captured in Tab. 1. Introducing ELENE signals improves the performance of GINs, and our single-run results EXP and SR25 are consistent with our formal analysis on §6—namely, ELENE and ELENE-L **(ND)** and **(ED)** all reach 100% accuracy on the 1-WL equivalent EXP task, as expected from Theo. 1. Furthermore, ELENE-L **(ED)** can distinguish all 3-WL equivalent SRGs in the challenging SR25 dataset—providing empirical evidence for Theo. 4.

On Graph Properties and Counting Substructures (2 runs averaged, as in Zhao et al. (2022)), a GIN + ELENE-L **(ND)** model consistently outperforms GIN-AK *without context encoding*. In Counting, both ELENE variants and GIN-AK$^+$ are outperformed by SUN, but GIN+ELENE matches or outperforms GIN on every task, showing that ELENE features are informative and can boost performance by themselves.

On both tasks, we find that GIN+ELENE-L **(ED)** performs poorly—outperforming GIN+ELENE but not our baselines. This might be caused by model over-parametrization, as six node and edge-level embedding matrices are learned for 3 and 6 layers on Counting Substructures and Graph Properties respectively[7]. Finally, on the Graph Properties tasks of `IsConnected` and `Diameter`, a GIN-AK$^+$ with ELENE-L outperforms state-of-the-art results—and interestingly a GIN with ELENE-L **(ND)** outperforms all existing baselines on the `IsConnected` task. This can be further improved by using a GIN-AK$^+$ with ELENE-L (ND).

---

[7]Weight sharing may help over-parametrization by learning a single structural representation, trading off expressivity.

Table 1: Expressivity benchmark results. In EXP and SR25, introducing Elene-L yields the best performance per task, shown in **underlined bold**. We highlight the *best-performing configurations from Elene variants on GIN in italics*, which we consistently observe in the Node-Centric (**ND**) configuration except for isomorphism tasks.

| Model | EXP (Acc.) | SR25 (Acc.) | Count. Substr. (MAE) | | | | Graph Prop. ($\log_{10}$(MAE)) | | |
|---|---|---|---|---|---|---|---|---|---|
| | | | Tri. | Tail Tri. | Star | 4-Cycle | IsCon. | Diam. | Radius |
| GIN | 50% | 6.67% | 0.357 | 0.253 | 0.023 | 0.231 | -1.914 | -3.356 | -4.823 |
| SUN[†,(Frasca et al., 2022)] | — | — | **0.008** | **0.008** | **0.006** | **0.011** | -2.065 | -3.674 | **-5.636** |
| GIN-AK[†,(Zhao et al., 2022)] | 100% | 6.67% | 0.093 | 0.075 | 0.017 | 0.073 | -1.993 | -3.757 | -5.010 |
| GIN-AK[+] | 100% | 6.67% | 0.011 | 0.010 | 0.016 | 0.011 | -2.512 | -3.917 | -5.260 |
| GIN+ELENE | 100% | 6.67% | 0.024 | 0.023 | 0.020 | 0.041 | -2.218 | -3.656 | -5.024 |
| GIN+ELENE-L (ND) | 100% | 6.67% | *0.012* | *0.015* | *0.014* | *0.016* | *-2.620* | *-3.815* | *-5.117* |
| GIN+ELENE-L (ED) | 100% | 100% | 0.023 | 0.023 | 0.017 | 0.023 | -2.497 | -3.541 | -4.755 |
| Best (GIN / GIN-AK) + (ELENE / ELENE-L) | 100% | 100% | 0.010 | 0.010 | 0.014 | 0.011 | **-2.715** | **-4.072** | -5.267 |

### 7.3 $h$-Proximity

We evaluate Elene-L on $h$-Proximity (Abboud et al., 2022) tasks (10-fold averaged)—where nodes are assigned colors including red and blue, and models classify whether all red nodes have at most two blue nodes within $h$ hops (positive) or otherwise (negative), as in Fig. 2. Models must learn which colors are relevant for the target and capture long-ranging dependencies during learning. Edge information is irrelevant, and pre-computed encodings like Elene cannot capture interactions of distances and node attributes.

In Abboud et al. (2022), the authors reported that MP-GNNs perform well on $h = 1$-Proximity, so we focus on the $h \in \{3, 5, 8, 10\}$ variants. Tab. 2 shows our results, where Elene-L (**ND**) outperforms strong baselines from SPNNs and Graphormer (Abboud et al., 2021). As expected, a GIN + Elene did not meaningfully improve over GIN. Our numerical results provide empirical validation for Theo. 5.

Table 2: $h$-Proximity binary classification results (accuracy). Elene-L **(ND)** without degree information outperforms baselines strong SPNNs and Graphormer baselines from[†] Abboud et al. (2021).

| | 3-Prox. | 5-Prox. | 8-Prox. | 10-Prox. |
|---|---|---|---|---|
| GCN[†] | $50.0 \pm 0.0$ | $50.0 \pm 0.0$ | $50.1 \pm 0.0$ | $49.9 \pm 0.0$ |
| GAT[†] | $50.4 \pm 1.0$ | $49.9 \pm 0.0$ | $50.0 \pm 0.0$ | $50.0 \pm 0.0$ |
| SPNN $(k = 1)$[†] | $50.5 \pm 0.7$ | $50.2 \pm 1.0$ | $50.0 \pm 0.9$ | $49.8 \pm 0.8$ |
| SPNN $(k = 5)$[†] | $95.5 \pm 1.6$ | $96.8 \pm 0.7$ | $96.8 \pm 0.6$ | $96.8 \pm 0.6$ |
| Graphormer[†] | $94.7 \pm 2.7$ | $95.1 \pm 1.8$ | $97.3 \pm 1.4$ | $96.8 \pm 2.1$ |
| GIN+ELENE | $52.0 \pm 2.0$ | $51.8 \pm 1.2$ | $52.4 \pm 2.6$ | $51.4 \pm 1.1$ |
| GIN+ELENE-L (ND) | $\mathbf{98.3 \pm 0.5}$ | $\mathbf{98.6 \pm 0.5}$ | $\mathbf{99.0 \pm 0.5}$ | $\mathbf{99.2 \pm 0.3}$ |

### 7.4 Real World Graphs

We also evaluate Elene and Elene-L on five real-world, large-scale graph classification and regression tasks. We test Elene and Elene-L on ZINC, MolHIV, PATTERN, CIFAR10, and MolPCBA and report our results in Tab. 3. Given increased memory and computation costs and the weaker performance of Elene-L (**ED**) in §7.2, we only evaluate Elene-L (**ND**).

On ZINC, GIN + Elene-L (3 averaged runs) achieves comparable results to existing baselines, including SUN (Frasca et al., 2022). Furthermore, by introducing Elene-L on GIN-AK[+], the model matches the previous strong baseline achieved by CIN (Bodnar et al., 2021). On PATTERN (3 averaged runs), GIN + Elene-L achieves comparable results to GIN-AK[+], but does not meet the best reported performance of

Table 3: Results on real world benchmark datasets. We compare with published results and reproduce the experiments of Zhao et al. (2022). Adding ELENE variants to GIN and GIN-AK$^+$ yield state-of-the-art results on ZINC and MolPCBA, and match the performance of existing methods in PATTERN and MolHIV.

| | ZINC (MAE) | PATTERN (Acc.) | MolHIV (ROC) | CIFAR10 (Acc.) | MolPCBA (AP) |
|---|---|---|---|---|---|
| **GSN**$^\dagger$ | $0.115 \pm 0.012$ | — | $77.99 \pm 1.00$ | — | — |
| **NGNN**$^\dagger$ | — | — | $78.34 \pm 1.86$ | — | $28.32 \pm 0.41$ |
| **CIN**$^\dagger$ | $0.079 \pm 0.006$ | — | $\mathbf{80.94 \pm 0.57}$ | — | — |
| **SUN**$^\dagger$ | $0.083 \pm 0.003$ | — | $80.55 \pm 0.55$ | — | — |
| **GCN-AK**$^{+\dagger}$ | $0.127 \pm 0.004$ | $\mathbf{86.887 \pm 0.009}$ | $79.28 \pm 1.01$ | $\mathbf{72.70 \pm 0.29}$ | $0.285 \pm 0.000$ |
| **GIN-AK**$^\dagger$ | $0.094 \pm 0.005$ | $86.803 \pm 0.044$ | $78.29 \pm 1.21$ | $67.51 \pm 0.21$ | $0.274 \pm 0.000$ |
| **GIN-AK**$^+$ | $0.082 \pm 0.003$ | $86.868 \pm 0.028$ | $77.37 \pm 0.31$ | $72.39 \pm 0.38$ | $0.293 \pm 0.004$ |
| (Lit. results$^\dagger$) | $^\dagger 0.080 \pm 0.001$ | $^\dagger 86.850 \pm 0.057$ | $^\dagger 79.61 \pm 1.19$ | $^\dagger 72.19 \pm 0.13$ | $^\dagger 0.293 \pm 0.004$ |
| **GIN** | $0.155 \pm 0.005$ | $85.692 \pm 0.042$ | $78.72 \pm 0.54$ | $59.55 \pm 0.54$ | $0.271 \pm 0.003$ |
| (Lit. results$^\dagger$) | $^\dagger 0.163 \pm 0.004$ | $^\dagger 85.732 \pm 0.023$ | $^\dagger 78.81 \pm 1.01$ | $^\dagger 59.82 \pm 0.33$ | $^\dagger 0.268 \pm 0.001$ |
| **GIN+IGEL** | $0.103 \pm 0.004$ | $86.762 \pm 0.029$ | $78.92 \pm 0.92$ | | |
| **GIN+ELENE** | $0.092 \pm 0.001$ | $86.783 \pm 0.044$ | $78.92 \pm 0.35$ | $56.34 \pm 0.06$ | $0.277 \pm 0.002$ |
| **GIN+ELENE-L (ND)** | $0.083 \pm 0.004$ | $86.828 \pm 0.002$ | $78.26 \pm 0.93$ | $68.95 \pm 0.25$ | $\mathbf{0.294 \pm 0.001}$ |
| **Best Result (ELENE / ELENE-L)** | $\mathbf{0.079 \pm 0.003}$ | $86.828 \pm 0.002$ | $79.15 \pm 1.45$ | $68.95 \pm 0.25$ | $\mathbf{0.294 \pm 0.001}$ |

GCN-AK$^+$ by a 0.07% delta. We do not achieve to independently reproduce GIN-AK$^+$ results (Zhao et al., 2022) on MolHIV (5 averaged runs)—finding that GIN with ELENE or ELENE-L do not have statistically significant ($p < 0.01$) differences with GIN, while the performance of GIN-AK$^+$ is statistically inferior.

Tab. 4 shows time and memory costs of ELENE compared to state-of-the-art methods. Despite not tuning hyperparameters, a GIN+ELENE-L model outperforms GIN-AK in CIFAR and a strong GIN-AK$^+$ baseline in MolPCBA. Furthermore, our setup of GIN layers combined with ELENE always outperforms GNN-AK$^+$ in terms of memory consumption. In ZINC, GIN+ELENE-L (**ND**) requires 0.99GB compared to the 1.68GB of GIN-AK$^+$ while reaching comparable performance ($0.083 \pm 0.004$ vs $0.082 \pm 0.003$, respectively). In MolHIV, GIN+ELENE model requires only 70MB during training while outperforming the ROC of our reproduced run of GIN-AK$^+$, which required 790MB—an 11.3-fold reduction in memory usage. On PATTERN, we find that GIN+ELENE-L (**ND**) achieves 99.95% of the performance of GIN-AK$^+$ while consuming only 7.8GB of memory during training, compared to the 26.52GB reported by Zhao et al. (2022)— a 3.4-fold reduction.

In summary, ELENE and ELENE-L (**ND**) achieve comparable results to the baselines with favorable time / memory efficiency. ELENE encodings used as node-features (**GIN+ELENE**) add minor overhead over GIN and match or outperform **GIN+IGEL** in all tested settings, and GIN-AK in four over five. ELENE-L (**ND**) also shows favorable memory performance versus GIN-AK and GIN-AK$^+$ in all setups. Finally, we observe additional memory costs for ELENE-L (**ED**) due to using node and edge embeddings.

Table 4: Memory and time performance on benchmark datasets, controlling for shared resource use as per §7.1. We report average epoch duration in seconds (s) and maximum memory consumption in gigabytes (GB) respectively. Dashed entries indicate executions that terminated due to running out of memory.

| | ZINC | | PATTERN | | MolHIV | | CIFAR10 | | MolPCBA | |
|---|---|---|---|---|---|---|---|---|---|---|
| **GIN** | 6.02s | 0.12GB | 118.62s | 1.42GB | 14.88s | 0.07GB | 98.37s | 0.90GB | 223.13s | 0.44GB |
| **GIN-AK** | 9.76s | 1.11GB | — | — | 19.30s | 0.64GB | 283.93s | 18.80GB | 534.78s | 3.80GB |
| **GIN-AK**$^+$ | 13.63s | 1.68GB | — | — | 25.47s | 0.79GB | — | — | 607.89s | 3.83GB |
| **GIN+ELENE** | 6.14s | 0.13GB | 90.15s | 1.47GB | 14.94s | 0.07GB | 120.21s | 0.91GB | 278.29s | 0.46GB |
| **+ELENE-L (ND)** | 10.23s | 0.99GB | 146.15s | 7.80GB | 42.53s | 0.54GB | 224.43s | 10.72GB | 451.12s | 2.39GB |
| **+ELENE-L (ED)** | 22.61s | 2.85GB | — | — | 32.28s | 1.40GB | — | — | 1025.58s | 7.10GB |

## 7.5 Memory Scalability

Finally, we evaluate the scalability of ELENE-L in a learning setting. We analyze the memory consumption as a function of the graph size and how it compares with other methods. For that, we follow a similar setting as Mitton & Murray-Smith (2023) for SPEN: we design and implement a learning task on a large $d$-regular graph, and use it to explore memory consumption under different values of degree $d$ and nodes $n$. With this setup, we train the model for 25 epochs to predict a constant variable so that both input tensors and gradient computations are kept in memory.

We evaluate both ELENE-L (**ND**) and (**ED**), together with a GIN model without any ELENE-L features, GIN-AK and GIN-AK$^+$ as baselines. For all ELENE-L variants, we execute the benchmark with different values of $k \in \{1, 2, 3\}$.

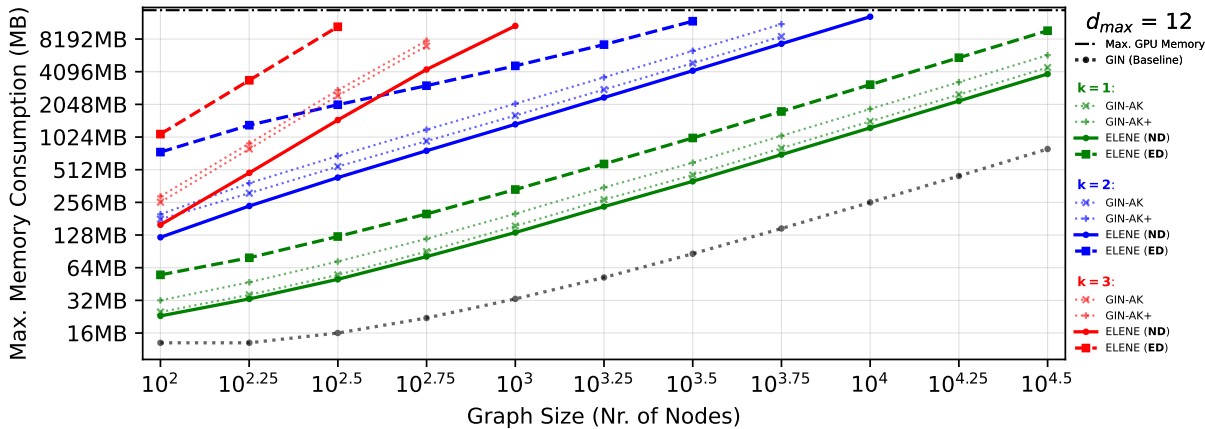

Figure 7: Memory scalability analysis of ELENE when $d_{\mathtt{max}} = 12$. We include GIN (dotted line) and maximum GPU memory (dash-and-dotted line) as indicative lower and upper memory bounds. ELENE-L (**ND**, full lines) outperforms both GIN-AK, GIN-AK$^+$ (dotted lines) and ELENE-L (**ED**, dashed lines). Additionally, ELENE-L (**ND**) can encode all $d$-regular graphs in the benchmark when $k = 1$. As expected, memory consumption increases linearly with the number of nodes as $d_{\mathtt{max}}$ is kept fixed.

Fig. 7 shows our results when $d_{\mathtt{max}} = 12$. As expected, the memory cost grows exponentially as a function of $n$. We observe that ELENE-L (**ND**) with $k = 2$ can scale up to graphs with $10,000$ nodes with ego-network sub-graphs. Since all nodes have the same degree $d_{\mathtt{max}} = 12$, at 2 hops we are guaranteed to find the root node, its 12 neighbors, and at least one additional neighbor at 2 hops —or 14 total nodes. In practice, as the graphs are randomly generated, we find that each of the 2-depth subgraphs contains an average of 144.13 nodes with the expected maximum at 145.

Additionally, our experiments show that ELENE-L (**ND**) with $k = 1$ can scale up to graphs with $10^{4.5} = 31,623$ nodes with up to $d_{\mathtt{max}} = 18$. Furthermore, despite requiring additional memory, the more expressive ELENE-L (**ED**) can nevertheless be used for $k = 1$ for graphs with up to $10^{4.5}$ nodes as well. Scaling to graphs with more than $10^5$ nodes is possible by increasing the cpu-memory (our limit is 96GB) or alternatively changing the implementation. The latter can be done for example by computing the encodings through parallel BFS, which would result in a slower algorithm (but of the same order of complexity). We provide additional results for $d_{\mathtt{max}} = 6$ and $d_{\mathtt{max}} = 18$, as well as different graph density patterns, in §B.2.

Recent methods like SPEN outperform global permutation-equivariant methods like PPGN. However, SPEN still struggles to process significantly smaller graphs, with $n \approx 1,000$ nodes and $k = 1$ depth ego-networks that contain 9 nodes, even with higher GPU memory as reported by Mitton & Murray-Smith (2023). Compared with the complexity of SPEN, which is $\mathcal{O}(n \cdot |\mathcal{V}_v^k|^2)$, ELENE-L can encode ego-networks with 16 times more nodes with comparable memory usage—while outperforming sub-graph GNN baselines like GIN-AK and GIN-AK$^+$.

## 7.6 Experiments Summary

ELENE and ELENE-L consistently boost GNN performance on the three experimental benchmarks, and are shown to be scalable in Tab. 4 and Fig. 7.

On Expressivity, §7.2 gives empirical support for Theo. 4, i.e. that ELENE-L (**ED**) can distinguish SRGs, *achieving 100% accuracy* on the challenging SR25 (Balcilar et al., 2021) dataset. Although SUN (Frasca et al., 2022) outperforms other models on Counting Substructures, ELENE and ELENE-L still improve baseline performance and match previous GIN-AK and GIN-AK$^+$ baselines respectively. On Graph Properties, GIN+ELENE-L matches existing baselines, and a GIN-AK$^+$ model with ELENE-L *outperforms* previous state-of-the-art results on the `IsCon.` and `Diam.` tasks with -2.715 and -4.072 $\log_{10}$(MSE) each.

On $h$-Proximity, §7.3 validates Theo. 5, i.e., that ELENE-L (**ND**) is at least as expressive as SPNNs (Abboud et al., 2022), as ELENE-L (**ND**) *outperforms SPNNs and Graphormers* at capturing *attributed structures—* that sparse ELENE vectors alone *cannot capture.*

On Real World Graphs from §7.4, ELENE and ELENE-L reach state-of-the-art results. On ZINC, GIN-AK$^+$ with ELENE-L achieves $0.079 \pm 0.003$ MAE, matching CIN (Bodnar et al., 2021). A GIN+ELENE-L matches 99.95% of the performance of baselines on PATTERN while *consuming 3.4× less memory*, and GIN+ELENE reaches 0.1% less accuracy than GIN-AK$^+$ but does so using 1.47GB, compared to the 26.54GB reported for GIN-AK$^+$—*a 18.1× memory reduction*. Finally, a GIN+ELENE-L *matches state-of-the-art results on MolPCBA* $(0.294 \pm 0.001$ vs $0.293 \pm 0.003$ of GIN-AK$^+$ (Zhao et al., 2022)) without hyper-parameter tuning and while *consuming 37.60% less memory (2.39GB vs 3.83GB).*

On Memory Scalability, §7.5 shows that ELENE-L can be used on $d$-regular graphs with more than $10^4$ nodes where $d_{\texttt{max}} \in \{6, 12, 18\}$, validating the expected memory costs from §3.3 and outperforming the memory consumption of strong GIN-AK and GIN-AK$^+$ baselines and recent methods like SPEN (Mitton & Murray-Smith, 2023).

## 8 Conclusions

We presented ELENE, a principled edge-level ego-network encoding capturing the structural signals sufficient to distinguish 3-WL equivalent SRGs. We proposed two variants—ELENE and ELENE-L—and showed that Node-Centric and Edge-Centric representations exhibit different expressive power. To position our findings, we formally drew connections between ELENE and recent Sub-Graph GNNs, Graph Transformers, and Shortest Path Neural Networks.

Empirically, we evaluated our methods on 10 different tasks, where the sparse ELENE vectors improve performance on structural expressivity tasks. Our learnable Edge-Centric ELENE-L variant boosts MP-GNN expressivity to reach 100% accuracy on the challenging SR25 dataset, while its Node-Centric counterpart improves over a strong baseline on the $h$-Proximity task and matches state-of-the-art results in several real-world graphs. Finally, we found our methods provide a trade-off between memory usage and structural expressivity, improving memory usage with up to 18.1× lower memory costs compared to sub-graph GNN baselines.

### Broader Impact Statement

Our main contributions are (a) a novel family of edge-aware features that can be used alone or during learning in MP-GNNs, with (b) a formal analysis of their expressivity that shows they can distinguish challenging SRGs, and (c) experimental results matching state-of-the-art learning models with favorable memory costs.

We do not foresee ethical implications of our theoretical findings. Our experimental results are competitive with state-of-the-art methods at a lower memory footprint, which may help solve tasks with limited memory budgets.

**Acknowledgements**

This work is part of the action CNS2022-136178 financed by MCIN/AEI/10.13039/501100011033 and by the EU Next Generation EU/PRTR. This work has been co-funded by MCIN/AEI/10.13039/501100011033 under the Maria de Maeztu Units of Excellence Programme (CEX2021-001195-M).

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

## A ELENE through BFS

This appendix showcases a BFS implementation of the ELENE Encoding that spans edges until reaching the maximum encoding depth $k$ given a node $v$ in $V$. As noted in §3.3, this implementation can be trivially parallelized over $p$ processors as the encoding of each $v \in V$ is independent of other nodes.

---

**Algorithm 1** ELENE Node Encoding using BFS.

---

**Input:** $G = (V, E), v \in V, k : \mathbb{N}$

1: `distances := {v : 0}`                                      ▷ Mapping of nodes to their distance to $n$, i.e. $l_G(u, v)$.
2: `r_degrees := {v : (0, 0, 0)}`                        ▷ Mapping of nodes to their relative degrees, i.e. $d_{\mathcal{S}}^{(p)}(u|v)$.
3: **for** `(src, dst)` in $G$.`bfs_edges`$(n, $`max_depth` $= k)$ **do**
4:     **if** `src` $\notin$ `distances` **then**          ▷ Invariant: only one node can be unknown / not in `distances`.
5:         `dst, src := src, dst`
6:     **end if**
7:     **if** `dst` $\notin$ `distances` **then**                          ▷ `dst` is unknown, so its distance is one-hop after `src`'s.
8:         `distances[dst] := distances[src]`
9:     **end if**
10:     `dist_delta := distances[dst] − distances[src]`          ▷ Compute the distance delta in {-1, 0, 1}.
11:
12:     ▷ Access the relative degree counts of each node.
13:     `src_deg := r_degrees.get(src, [0, 0, 0])`
14:     `dst_deg := r_degrees.get(dst, [0, 0, 0])`
15:
16:     ▷ Increment degree counts for each node in their respective 'direction'.
17:     `src_deg[dist_delta + 1]++`        ▷ The indexing maps {-1, 0, 1} deltas into {0, 1, 2} vector indexes.
18:     `dst_deg[1 − dist_delta]++`
19:
20:     ▷ Update the relative degrees of `src` and `dst`.
21:     `r_degrees[src] := src_deg`
22:     `r_degrees[dst] := dst_deg`
23: **end for**
24:
25: ▷ For each $u \in \mathcal{V}_v^k$, compute Eq. 1 quadruplets, count their frequencies and return the mapping.
26: `mapping := {}`
27: **for** $u \in \mathcal{V}_v^k$ **do**
28:     `quadruplet := (distances[u], r_degrees[u][0], r_degrees[u].sum(), r_degrees[u][2])`
29:     `mapping[quadruplet]++`
30: **end for**
**Output:** `mapping`

---

## B Benchmark Details

In this appendix, we provide an overview of the benchmark we execute when evaluating ELENE, including the variants of models we test, and descriptions of our code and compute environment. We also summarize the datasets we use in §B.1.

**Benchmark Configuration.** We build on top of the implementation from Zhao et al. (2022), introducing explicit ego-network attributes on their evaluation framework for consistency.

All ELENE results are reported by extending the node and edge attributes as input into a GIN Xu et al. (2019) extended to support edge-level features when available Hu et al. (2020b). In all experiments, we evaluate ELENE-L on top of GINs with edge extensions Hu et al. (2020b).

For all explicit ego-network attribute methods, we summarize the available hyper-parameters in Tab. 5. For the implementation of Elene-L, we observed unstable training when using the sum pooling function during early stages of development. We found that training was stable using masked `Mean` pooling where the $n$ node messages (or $m$ for edge messages) in the ego-network sub-graph are averaged considering a binary mask for neighbors of the root node at a distance $k$ or less. All our results are reported using `Mean` pooling, including our results on SR25, suggesting that this decision does not adversely impact the model expressivity expected from §6. The resulting implementation of Eq. 4 is:

$$\Phi_{\mathtt{ND}}^t(v) = \Phi_{\mathtt{out}}\left(\mathbf{x}_v^t \Bigg\| \sum_u^{\mathcal{V}_v^k} \frac{\Phi_{\mathtt{nd}}^t(u|v)}{\mathtt{size}(\mathcal{V}_v^k)} \Bigg\| \sum_{\langle u,w \rangle}^{\mathcal{E}_v^k} \frac{\Phi_{\mathtt{ed}}^t(u,w|v)}{\mathtt{size}(\mathcal{E}_v^k)}\right).$$

We use an analogous implementation for Eq. 7. Additionally, in our experimental benchmark we choose to implement Elene-L (**ND**) without the $\Phi_{\mathtt{ed}}^t$ term so that it more closely follows the node-centric Elene encodings of Eq. 1, reducing memory costs. We describe the hyper-parameters implemented to control our models in Tab. 5.

Table 5: Hyper-parameters controlling the behaviour of explicit ego-network attribute encodings. Elene only relies on $k$, while Elene-L has 5 additional configurable settings.

| Parameter | Elene | Elene-L |
|---|---|---|
| **Depth of Ego-Net** ($k$) | {0, 1, 2} | {0, 1, 2, 3} |
| **Embedding Type** | Sparse | Dense, learned |
| **Representation** | Node-only | Node-centric (**ND**), Edge-centric (**ED**) |
| **Max. Encoded Degree** | Set to $d_{\mathtt{max}}$ from the training dataset. | Set to $d_{\mathtt{max}}$ from the training dataset or 0 (ignore degree info). |
| **Max. Encoded Distance** | Equal to $k$ | Set to $k$. Can be modified to control the sub-graph mean norm. factor. |

**Tested Models.** On the Expressivity tasks, ZINC and MolHIV, we evaluate all learnable variants (**ND** and **ED**), while on the remaining classification/regression benchmarks we only consider (**ND**) models due to reduced memory costs and limited computational bandwidth. Furthermore, in all Elene-L setups we only test a reduced number of hyper-parameters due to computational constraints, unless specified otherwise, only evaluating different values of the maximum sub-graph distance to embed. We describe the hyper-parameters and modeling choices in detail in §B.2.

## B.1 Dataset Details

We summarize the key aspects of the datasets we use to evaluate our proposed methods in §7. Tab. 6 contains an overview of each benchmark and dataset, the objective being addressed, and high-level dataset statistics—namely number of graphs, average number of nodes ($n$) and edges ($m$) per graph.

## B.2 Detailed Experimental Summary

In this section, we summarize our experimental setup and training procedure, describing the hyper-parameters that we consider in each setting. For all the experiments described, we evaluate the Elene encodings by concatenating them with the node feature vectors and as part of the edge features when available, using the element-wise product following the same approach as Igel.

**Expressivity.** See `expressivityDatasets.sh` for details.

Table 6: Dataset statistics.

| Benchmark | Dataset | Objective | Tasks | Nr. of Graphs (Train / Valid / Test) | Avg. $n$ | Avg. $m$ |
|---|---|---|---|---|---|---|
| Expressivity | **EXP** | Distinguish 1-WL Equiv. graphs | 2 | 1200 | 44.4 | 110.2 |
| | **SR25** | Distinguish 3-WL Equiv. graphs | 15 | 15 | 25 | 300 |
| | **CountingSub.** | Count graph substructures | 4 | 1500 / 1000 / 2500 | 18.8 | 62.6 |
| | **GraphProp.** | Regress graph properties | 3 | 5120 / 640 / 1280 | 19.5 | 101.1 |
| Real World Graphs | **ZINC-12K** | Molecular prop. regression | 1 | 10000 / 1000 / 1000 | 23.1 | 49.8 |
| | **CIFAR10** | Multi-class class. | 10 | 45000 / 5000 / 10000 | 117.6 | 1129.8 |
| | **PATTERN** | Recognize subgraphs | 2 | 10000 / 2000 / 2000 | 118.9 | 6079.8 |
| | **MolHIV** | Binary class. | 1 | 32901 / 4113 / 4113 | 25.5 | 54.1 |
| | **MolPCBA** | Multi-label binary class. | 128 | 350343 / 43793 / 43793 | 25.6 | 55.4 |
| Proximity | $h$-**Proximity** | Binary classification | 4 | 9000 | 117.14 | 1484.82 |

—*EXP and SR25.* We evaluate ELENE on GIN and GIN-AK$^+$ for both data sets with $k \in \{0, 1, 2\}$. For ELENE-L, we evaluate all model variants for $k \in \{0, 1, 2\}$ with 8-dim embeddings for EXP and 32-dim embeddings for SR25. All models use $L = 4$ for EXP and $L = 2$ for SR25.

—*Counting Sub. and Graph Prop.* We evaluate ELENE on GIN and GIN-AK$^+$ for both data sets with $k \in \{0, 1, 2\}$. For ELENE-L, we evaluate all model variants for $k \in \{0, 1, 2\}$ with 16-dim embeddings. On the GraphProp dataset, we additionally try $k = 3$ after noticing expected positive results during early evaluation—as larger values of $k$ enable the model to capture long-range dependencies. All models use $L = 3$ for Counting Sub. and $L = 6$ for Graph Prop.

**Real World Graphs.** See `benchmarkDatasets.sh` for details.

—*ZINC and MolHIV.* We evaluate ELENE on GIN and GIN-AK$^+$ for both data sets with $k \in \{0, 1, 2\}$. For ELENE-L, we evaluate all model variants for $k \in \{0, 1, 2, 3\}$ with 32-dim embeddings. All models use $L = 6$ for ZINC and $L = 2$ for MolHIV.

—*PATTERN.* We evaluate ELENE on GIN with $k \in \{0, 1, 2\}$ and on GIN-AK$^+$ $k \in \{0, 1\}$. For ELENE-L, we evaluate all model variants for $k \in \{0, 1, 2, 3\}$ with 64-dim embeddings. Suspecting that degree information may not play a salient role in the sub-graph patterns, we also evaluate the setting without degree information but found this slightly degrades performance compared to models that encode degree attributes. All models use $L = 6$.

—*CIFAR10 and MolPCBA.* We evaluate node-centric ELENE-L (**ND**) with $k \in \{1, 2, 3\}$. Due to computational constraints, we prioritize training with $k = 3$ given promising results in other tasks. On CIFAR, we discard uninformative degree information as graphs are $k = 8$ nearest neighbor graphs containing super-pixel information. We do not modify the architecture or hyper-parameters of the best-performing GNN-AK$^+$ model reported in Zhao et al. (2022). Our results report average and standard deviations of the evaluation metric—Accuracy for CIFAR10, Average Precision (AP) for MolPCBA—collected from 3 independent runs.

$h$-**Proximity.** See `proximityResults.sh` for details.

We evaluate node-centric ELENE-L without degree information, which matches the configuration of SPNNs. We do not tune any hyper-parameters, evaluating ELENE-L with $k \in \{3, 5\}$ fixing $L = 3$ and using 32-dim. embeddings. The first layer in the network embeds the color information, for which the model needs to appropriately learn to ignore irrelevant colors. Due to constrained computational resources, we only evaluate two maximum distances for ELENE-L, 3 and 5, sharing embedding weights and introducing ego-network signals before each of the 3 GIN layers. We share ELENE-L embedding matrices across all layers and set the maximum encoded degree $d = 0$ to only encode distance information. We report the mean and standard deviation of the binary classification accuracy computed across 10-folds over the dataset, following Abboud et al. (2022).

**Memory Scalability.**

We provide additional results from the memory scalability experiments in §7.5, reporting memory consumption performance when $d_{\mathtt{max}} = 6$ and $d_{\mathtt{max}} = 18$ in Fig. 8 and Fig. 9.

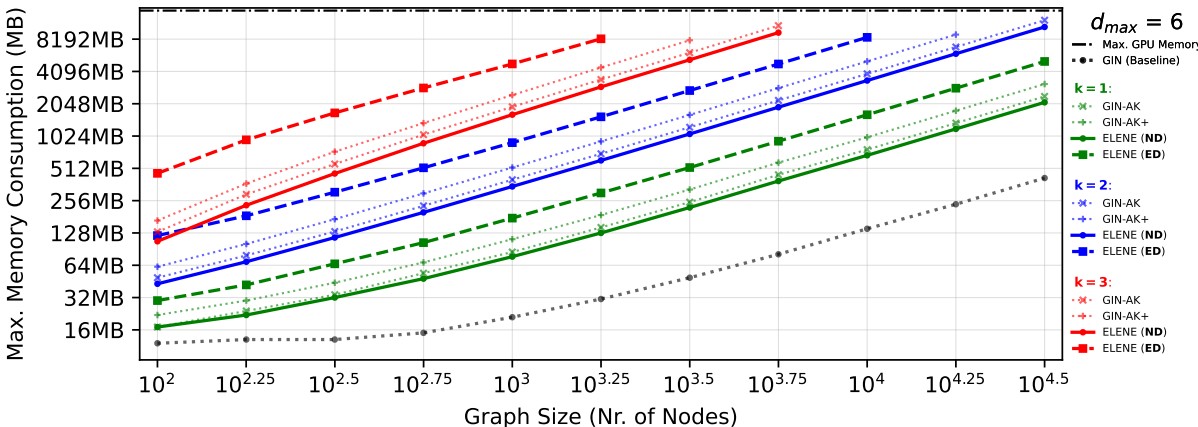

Figure 8: Memory scalability analysis of ELENE with $d_{\mathtt{max}} = 6$, produced as Fig. 7. We include GIN (dotted line) and maximum GPU memory (dash-and-dotted line) as indicative lower and upper memory bounds. ELENE-L (**ND**, full lines) outperforms both GIN-AK, GIN-AK$^+$ (dotted lines) and ELENE-L (**ED**, dashed lines). Additionally, ELENE-L (**ND**) can encode all $d$-regular graphs in the benchmark when $k = 1$. As expected, memory consumption increases linearly with the number of nodes as $d_{\mathtt{max}}$ is kept fixed.

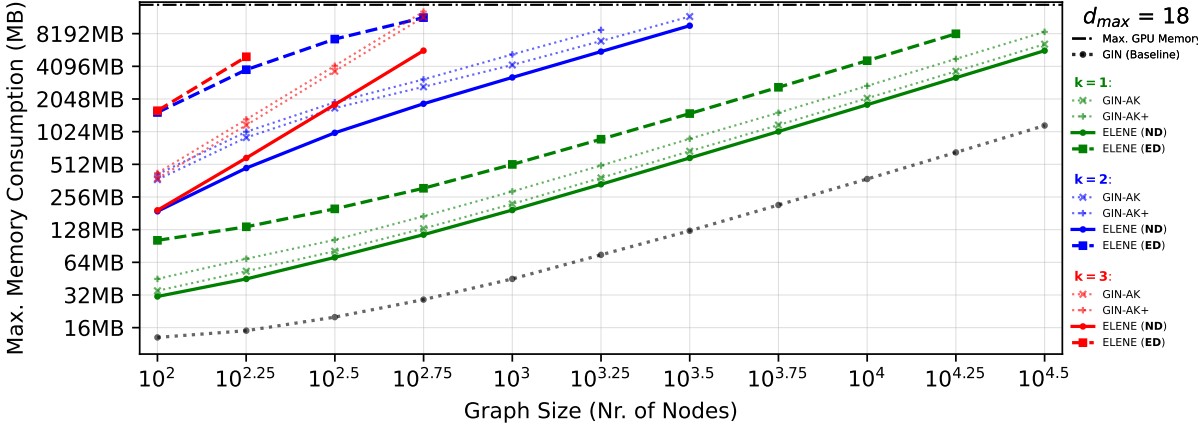

Figure 9: Memory scalability analysis of ELENE with $d_{\mathtt{max}} = 18$. See caption of Figure 8 for details.

**Graph Density and Scalability.**

We also provide extended memory scalability results by studying the impact of the density of the graph. We ran additional experiments on graphs where $N = 1000$, and evaluated the memory consumption as density increases as a function of the degree of nodes in the graph. We perform the same experiment in two settings: one where the degree distribution is regular (i.e., the graphs are $d$-regular, studying different values of $d$), and one where the distribution of degrees is irregular. In the irregular case, we study the case in which all nodes have *at least* degree $d$, but may have higher connectivity following the Barabási-Albert preferential attachment model (Barabási & Albert, 1999).

*— Memory Consumption on Regular Density Graphs.* In Fig. 10, we compare GIN, GIN-AK, GIN-AK$^+$ and ELENE variants on at depths $k \in \{1, 2, 3\}$. Note that we could not include SPEN, as described in §7.5, due to reaching the maximum memory thresholds at $d_{\max} = 8$.

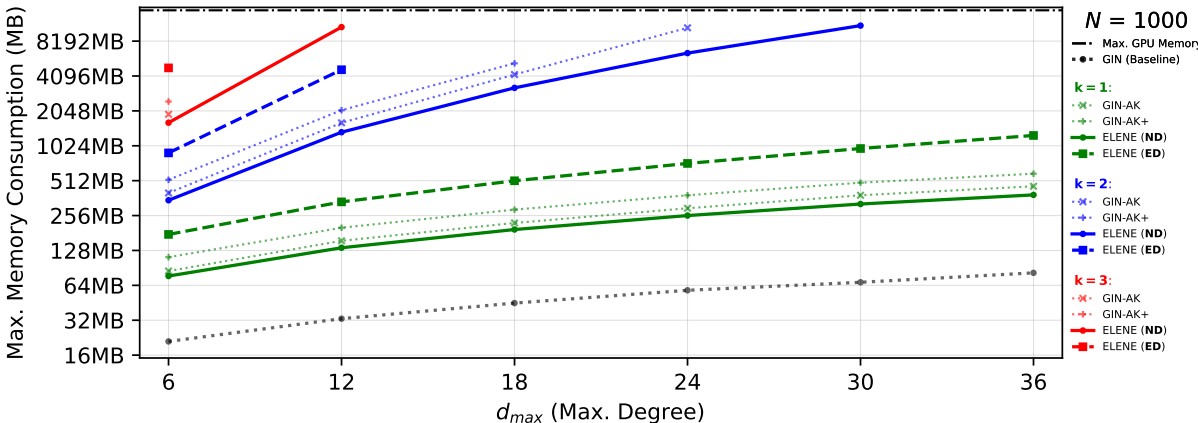

Figure 10: Memory scalability analysis of ELENE with $N = 1000$ in function of increasing values of $d_{\max}$. See caption of Figure 8 for details.

*— Memory Consumption on Irregular Density Graphs.* In Fig. 11, we repeat the analysis from Fig. 10 on graphs generated following the preferential attachment model where each node has at least $m$ edges. We find that ELENE-L (**ND**, full lines) outperforms both GIN-AK, GIN-AK$^+$ (dotted lines) and ELENE-L (**ED**, dashed lines), matching §7.5 and results on regular connectivity patterns shown in Fig. 10.

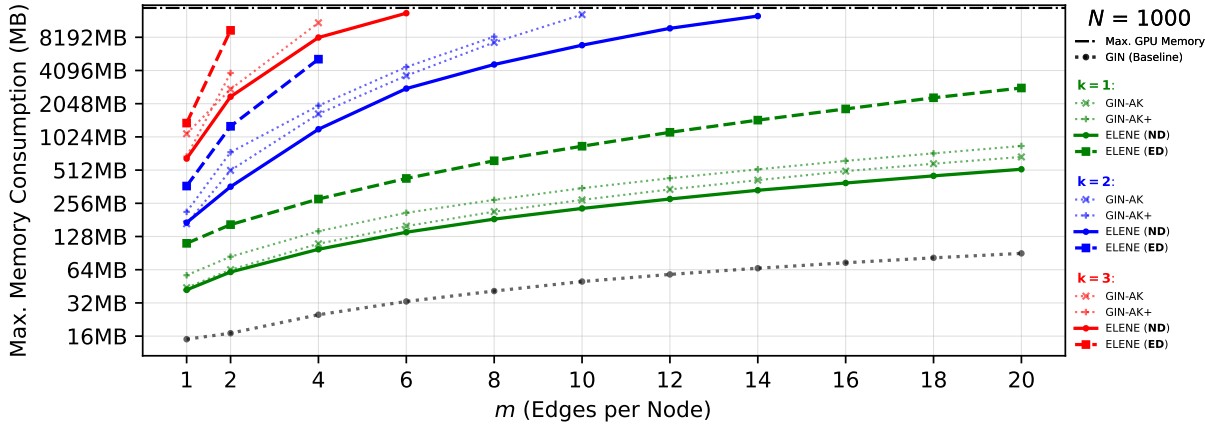

Figure 11: Memory scalability analysis of ELENE with $N = 1000$ in function of increasing values of $d_{\min}$ on random Barabási-Albert graphs. See caption of Figure 8 for details.

### B.3 Best Hyper-parameters

In this section, we provide an overview of the best hyperparameters we find for ELENE and ELENE-L. For simplicity, we only report the best performing model, i.e., not distinguishing between enhancing a GIN or a GIN-AK$^+$ model. We group together hyper-parameters set at the dataset level (e.g. for the Counting Substructures or $h$-Proximity datasets), and report the hyper-parameters corresponding to the best models reported in §7. In our summary, we include the best-performing ego-network feature with (a) the ego-network depth — $k$, (b) the number of layers — $L$, and (c) the embedding layer size for ELENE-L.

Table 7: Best hyper-parameters for the for the top performing models after introducing explicit ego-network attributes as shown in §7. We report the hyper-parameters corresponding to the best-performing model by looking at the objective performance metric on each dataset, and resolve ties by selecting the model with the lowest memory footprint.

| Benchmark | Dataset | Task | Ego-Net Feature | $k$-hops | $L$-Layers | Emb. Size (ELENE-L) |
|---|---|---|---|---|---|---|
| Expr. | **EXP** | **All Ego-Net Features Reach 100% Accuracy** | | 1 | 4 | 32 |
| | **SR25** | **GIN +ELENE-L (ED)** | | 1 | 2 | 32 |
| | **Counting Sub.** | **Triangle** | **GIN-AK$^+$+ELENE** | 2 | 3 | 16 |
| | | **Tailed Tri.** | **GIN-AK$^+$+ELENE** | | | |
| | | **Star** | **GIN +ELENE-L (ND)** | | | |
| | | **4-Cycle** | **GIN-AK$^+$+ELENE** | | | |
| | **Graph Prop.** | **IsConn.** | **GIN-AK$^+$+ELENE-L (ND)** | 3 | 6 | 16 |
| | | **Diameter** | **GIN-AK$^+$+ELENE-L (ND)** | | | |
| | | **Radius** | **GIN-AK$^+$+ELENE-L (ND)** | | | |
| Real World Graphs | **ZINC-12K** | | **GIN+ELENE-L (ND)** | 3 | 6 | 32 |
| | **CIFAR10** | | **GIN+ELENE-L (ND)** | 2 | 4 | 64 |
| | **PATTERN** | | **GIN+ELENE-L (ND)** | 2 | 6 | 64 |
| | **MolHIV** | | **GIN+ELENE** | 2 | 2 | N/A |
| | **MolPCBA** | | **GIN+ELENE-L (ND)** | 3 | 5 | 64 |
| Proximity | $h$-**Proximity** | $h = 3$ | **GIN + ELENE-L (ND)** | 3 | 3 | 32 |
| | | $h = 5$ | | 5 | | |
| | | $h = 8$ | | 5 | | |
| | | $h = 10$ | | 5 | | |

We summarise our findings in Tab. 7. For datasets and tasks where multiple models achieve comparable performance (i.e. same performance metric with the reported significant digits), we break ties by reporting the model with the lowest memory footprint across the tie.

## C ELENE is Permutation Equivariant and Invariant

We show that ELENE is permutation equivariant at the graph level, and permutation invariant at the node level. As all operations that ELENE requires are permutation equivariant at the graph level, and permutation invariant at the node level, the same holds for ELENE representations.

**Lemma 1.** *Given any $v \in V$ for $G = (V, E)$ and given a permuted graph $G' = (V', E')$ of $G$ produced by a permutation of node labels $\pi : V \to V'$ such that $\forall v \in V \Leftrightarrow \pi(v) \in V'$, $\forall(u, v) \in E \Leftrightarrow (\pi(u), \pi(v)) \in E'$.*

*All ELENE representations are permutation equivariant at the graph level:*

$$\pi(\{\!\{e_{v_1}^k, \ldots, e_{v_n}^k\}\!\}) = \{\!\{e_{\pi(v_1)}^k, \ldots, e_{\pi(v_n)}^k\}\!\}.$$

*Furthermore, ELENE representations are permutation invariant at the node level:*

$$e_v^k = e_{\pi(v)}^k, \forall v \in V, \pi(v) \in V'.$$

*Proof.* Note that $e_v^k$ in Eq. 1 can be expressed in terms of $d_G^{(p)}(u|v)$ and $l_G(u, v)$. Both $l_G(\cdot, \cdot)$ and $d_G^{(p)}(\cdot|\cdot)$ are permutation invariant functions at the node level and equivariant at the graph level, as they rely on the distance between nodes, which will not change when permutation $\pi(\cdot)$ is applied. Thus, ELENE representations are permutation equivariant at the graph level, and permutation invariant at the node level. $\square$

