# OpenReview forum: "Improving Subgraph-GNNs via Edge-Level Ego-Network Encodings"
_TMLR — Accepted by TMLR_

### Review · Reviewer_suwS · 2024-01-04

**Summary Of Contributions:**

The authors propose ELENE and ELENE(L), two approaches to include subgraph information in MPNNs without the need to run subgraph GNNs. The authors thoroughly discuss the relation of their paper to others, and provide strong motivation for the proposed method and research direction.

The method is nicely analyzed to show the expressive power of ELENE, followed by a significant amount of experiments and computational complexity, and cost analyses.

**Audience:**

Yes

**Broader Impact Concerns:**

No concerns.

**Claims And Evidence:**

Yes

**Requested Changes:**

Overall, I think that the paper is in great condition and is highly relevant. It should be published after minor changes, as described in my review.

**Strengths And Weaknesses:**

Strengths:

- The paper is well written. It was easy for me to follow and understand the method.

- The authors provide many comparisons with other methods, both qualitatively and quantitatively.

- The method seems novel to me, and seems to be effective in practice.

- The authors provide many details about the implementation and experimental settings.

Weaknesses:

- In page 1 the authors claim "This flexibility extends the expressive power of MP-GNNs and their robustness,". Can you please refer to the robustness aspect?

- In page 5 the authors mention several subgraph GNNs method ("GNN-AK,
NGNNs, SUN, ESAN, or SPEN."), but their citation only comes several pages after. I would expect the citation to at least be in order of appearance in text. Thus I would recommend to include the references there as well.

- In section 4, the authors first discuss a simple ELENE approach, but to my understanding there is no equation that defines it. Can you please add an equation so it is easier to compare the two proposed methods?

- In equations 5 and 6, is gamma learned? or is it a hyperparameter?

- The authors mention the over-squashing problem several times in the paper. Is there anything theoretical or practical that can be said about ELENE with respect to this problem?

- In terms of "subgraph expresiveness without running subgraph gnns", the authors should cite and discuss "Efficiently Counting Substructures by Subgraph GNNs without Running GNN on Subgraphs"

- In terms of lightweight / efficient subgraph GNNs, the authors should cite and discuss "Ordered Subgraph Aggregation Networks", "MAG-GNN: Reinforcement Learning Boosted Graph Neural Network", and "Efficient Subgraph GNNs by Learning Effective Selection Policies‏".

- In terms of paths in GNNs, the authors discuss the shortest path MPNN, but I would appreciate it if the authors can also discuss the differences between works like "pathGCN: Learning General Graph Spatial Operators from Paths‏ ", and "Path neural networks: Expressive and accurate graph neural networks".

---

> ### Author Response · Authors · 2024-01-18
> **Response to Reviewer suwS**
>
> We thank the reviewer for the helpful comments and suggestions. We will address the weaknesses pointed out by the reviewer in the revised manuscript as follows:
>
> > Can you please refer to the robustness aspect?
>
> We do not explicitly study the impact of introducing noise or uncertainty in our model. For clarity, we will remove any reference to robustness in the related work section of the revised manuscript.
>
> > I would expect the citation to at least be in order of appearance in text.
>
> We agree and will add the citations following their order of appearance.
>
> > Can you please add an equation [describing the simple ELENE variant] so it is easier to compare the two proposed methods?
>
> We introduce later, in Eq., 9 the approach that converts the ELENE multi-set in Eq. 1 into a flattened sparse vector representation containing the frequencies of every ELENE tuple. We will move Eq. 9 before section 4 and adapt the revised manuscript accordingly.
>
> > In equations 5 and 6, is gamma learned?
>
> Gamma is a learned parameter, we will specify this in the text of the updated manuscript.
>
> > Is there anything theoretical or practical that can be said about ELENE with respect to over-squashing?
>
> Only indirectly, given the relation between ELENE and SPNNs described in Section 6.3, as SPNNs were proposed to alleviate over-squashing. However, we believe that studying the connection of the over-squashing problem with edge-level information, ELENE and its variants could be promising future work.
>
> > Suggested related work: ESC-GNN, OSAN, MAG-GNN, Policy-Learn and PathGNNs
>
> We thank the reviewer for the additional related work which will help position our findings. We will include their relation to our work in the revised version of our manuscript as follows:
> * The Efficient Substructure Counting GNN (ESC-GNN) is closely related to the simple frequency-based encoding of ELENE features shown in Equation 9. Our theoretical work and learnable variants can be seen as an abstraction over purely count-based structural signals which connect them to both k-WL and other models like sub-graph GNNs, IGEL or SPNNs.
> * OSAN provides an alternative variant of 1-WL and its expressive variants that executes 1-WL on all ordered k-vertex subgraphs and provides a GNN architecture based on this isomorphism test. This approach is related to ELENE and other sub-graph methods, but requires exponential running time to collect all ordered k-vertex subgraphs and in practice relies on sampling.
> * MAG-GNN and Policy-Learn focus on learning sub-graph selection policies, such that sub-graph information relevant to the target task is kept while ignoring irrelevant structures. We believe this work could be connected with ELENE by leveraging the edge-level information as features during the selection process.
> * We focused on SPNNs due to their connections with Graphormers. However, we believe that it might be possible to formally connect Path GNNs with edge-level information. In particular, as Path GNNs operate on subsets of all paths within a graph, the edge-level signals in Section 3 may be able to encode similar information for a given node by tracing the distance and degree information of edges at every hop.

---

### Review · Reviewer_o54i · 2024-01-04

**Summary Of Contributions:**

The authors introduce a novel edge-level ego-network encoding (ELENE) to enhance the learning capabilities of Message Passing Graph Neural Networks (MP-GNNs). ELENE improves expressivity by providing additional node and edge features and extends standard message-passing schemes. The encoding shows superiority in distinguishing a challenging class of 3-WL equivalent graphs, namely the Strongly Regular Graphs.

Their theorems seem to be correct, and provide expressivity results over ELENE, and ELENE-L configurations. The paper develops a thorough experimental framework for evaluating ELENE's contribution to graph learning tasks. Its empirical evaluation across various benchmarks shows that ELENE matches or exceeds existing baselines in tasks like graph classification and regression, with a significant reduction in memory usage.

**Audience:**

Yes

**Broader Impact Concerns:**

The ethical implications are not discussed. I do not see however any particular reason of ethical implications that would require an addition of a Broader Impact Statement.

**Claims And Evidence:**

Yes

**Requested Changes:**

Following my thoughts on *Weaknesses* section, I suggest that the following changes would further improve the paper:
1. Inclusion of an increasing density for SRGs plot, showcasing the correlation with respect to the density. Thus, the authors can further validate the computational complexity claims.
2. Inclusion of competing baselines in the scalability evaluation. Figures 7,8, and 9 show only the comparison with GIN.
3. Another interesting addition would be the experimentation with different graph types (with more diverse density patterns) to further enhance the scalability claims.

**Strengths And Weaknesses:**

**Strengths**
- Novel, and rather interesting approach to graph learning methods, incorporating edge-level ego-network embeddings.
- Theoretical results show improvement in terms of expressivity, matching, and surpassing expressivity results from IGEL.
- *Technical validity*: The theorems seem correct, supporting the paper's arguments.
- One of the most interesting and selling points of the method is their efficient memory usage. In particular, they seem to be superior with respect to another competitive baseline GIN-AK$^+$.

**Weaknesses**
- [Density analysis] Section 7.5 could benefit from a more detailed investigation of denser graphs. A plot similar to 7, but with increasing density would be very insightful to capture the impact of the parameter $d$ of the SRGs.
- [Baseline Comparison] Since scalability is one of the main contributing axes of the method, it would be very helpful to report, also, the behavior of competing baselines with respect to increasing graph sizes (in Section 7.5).
- In Table 3, baselines (CIN and GCN-AK$^+$ seem to outperform ELENE on 3 out of 5 tasks. However, I do not see that as a strong weakness, since the required memory consumption is higher for the baselines.

---

> ### Author Response · Authors · 2024-01-18
> **Response to reviewer o54i**
>
> We thank the reviewer for the helpful comments and suggestions. We will address the weaknesses pointed out by the reviewer in the revised manuscript as follows:
>
> > [Density Analysis] Inclusion of an increasing density for SRGs plot, showcasing the correlation with respect to the density. Thus, the authors can further validate the computational complexity claims.
>
> We ran additional experiments using a fixed value of $N= 1000$ to evaluate the memory consumption as density increases as a function of $d_{\max}$ on d-regular graphs. We compare GIN, GIN-AK, GIN-AK$^{+}$ and ELENE variants at depths $k \in \{1, 2, 3\}$. Note that we could not include SPEN, as described in the paper, due to reaching the maximum memory thresholds at $d_{\max} = 8$. The results show the same tendency as the figure in the paper and are consistent with Table 4. We will include this figure in the next version of our manuscript.
>
> > [Baseline Comparison] Inclusion of competing baselines in the scalability evaluation. Figures 7,8, and 9 show only the comparison with GIN.
>
> We will generate additional figures, replacing Figures 7, 8, and 9 in the revised version, that include the memory consumption of GIN-AK and GIN-AK$^{+}$, which are the main approaches we compare against. As expected, we observed that these methods require more memory than ELENE (ND) but less than ELENE (ED).
>
> > Another interesting addition would be the experimentation with different graph types (with more diverse density patterns) to further enhance the scalability claims.
>
> We agree with the reviewer that exploring irregular density patterns would be valuable. We ran additional experiments on a variant of the memory benchmark similar to the plot for Density Analysis discussed above. Rather than d-regular graphs, we measure memory consumption in random graphs generated with the Barabási-Albert preferential attachment model. This lets us explore the irregular density case, where all nodes are guaranteed to have a minimum degree m, but certain nodes ‘dominate’ the network and appear in most sub-graphs when $k > 1$. The results show the same tendency as the figure in the paper and are consistent with Table 4. We will include this figure in the next version of our manuscript.

---

### Review · Reviewer_ddSg · 2024-02-27

**Summary Of Contributions:**

The paper proposes a novel edge-level ego-network encoding for learning on graphs, called ELENE, which can enhance the expressivity and performance of message-passing graph neural networks (MP-GNNs). ELENE captures the structural information of the two ego networks of adjacent nodes in the input graph and can distinguish strongly regular graphs, a family of hard-to-differentiate graphs. The paper also introduces two variants of ELENE, one that is sparse and one that is learnable. It shows that they have different expressive power depending on whether they are node-centric or edge-centric. The paper evaluates ELENE and its variants on various graph learning tasks, such as graph classification, graph regression, and proximity tasks. The results show that they significantly match or improve the state-of-the-art methods while reducing memory usage.

**Audience:**

Yes

**Broader Impact Concerns:**

The broader impact is fully discussed in the Broader Impact Statement section from my perspective.

**Claims And Evidence:**

Yes

**Requested Changes:**

Please kindly refer to the Weaknesses.

**Strengths And Weaknesses:**

Strengths:

* The paper addresses an important problem of enhancing the expressivity and performance of graph learning, which is a challenging and practical task in machine learning. The paper is genuinely well-written with helpful illustration figures.
* A novel and effective encoding of ELENE, which leverages the edge-level ego-network information to capture the structural signals of the input graph, is proposed to distinguish strongly regular graphs, a family of 3-WL equivalent graphs with theoretical groundings.
* Extensive experiments and analysis demonstrate the advantages of ELENE and ELENE-L over existing methods on various datasets and tasks. The paper also shows that ELENE and ELENE-L improve memory usage by up to 18.3x compared to sub-graph GNN baselines.

Weaknesses:
* As pointed out by the authors, approaches [1-2] similar to ELENE have been proposed recently, but the difference between ELENE and [1-2] is not explicitly discussed. This is essential especially since the theoretical findings are built upon the results from [1]. Therefore, a detailed discussion between ELENE and [1-2] as well as the experimental comparison between them is needed to further clarify the motivation.

* While Section 7.5 indicates the memory scalability of ELENE against GIN and GIN+AK, it is unclear whether ELENE could tackle graphs with massive nodes (more than 10^5). This concern arises since ELENE-L (ED) can hardly be employed on the reported real-world benchmark datasets (as indicated by the authors from Section 7.4). This might be available by introducing some large graph sampling techniques.

* The performance of ELENE achieves incremental improvement over a small set of baselines. For example, in Table 3, ELENE could only achieve on-par performance on two datasets and weaker performance on the other three in comparison with the other baselines. Therefore, the evaluation could be more convincing by introducing more related baselines (such as [1-2]) or other real-world benchmarks that could better demonstrate the expressivity like the ones in Table 1. In other words, how the enhancement of expressivity could be reflected in real-world settings needs further demonstration.

* In the discussion of memory usage comparison, it seems unfair to emphasize the comparison between GIN+ELENE and GIN-AK since the parameter complexity is different. From Table 4, it can be noticed that GIN+ELENE-L (ND) has comparable memory consumption against GIN-AK+ while GIN+ELENE-L (ED) uses more memory than GIN-AK+. Therefore, it would be better to conduct parameter complexity analysis to better reflect the memory usage analysis.

* Minor issues: [2] is not correctly cited in its published journal. What is the difference between GIN-AK and GIN-AK+?


[1] Beyond 1-WL with Local Ego-Network Encodings, LoG 22

[2] Improving Graph Neural Network Expressivity via Subgraph Isomorphism Counting, TPAMI 22

---

> ### Author Response · Authors · 2024-03-03
> **Response to Reviewer ddSg.**
>
> We thank the reviewer for the helpful comments and suggestions. We addressed the weaknesses pointed out by the reviewer in the revised manuscript as follows:
>
> > Differences with approaches [1-2]: detailed discussion & experimental comparisons are needed.
>
> IGEL [1] vectors contain counts of (distance, degree) tuples within ego-network sub-graphs but are edge-agnostic. That is, they skip the $d^1_u$ and $d^3_u$ components that encode relative degrees in Eq. 1. Therefore, IGEL vectors are a strict subset of the information in ELENE as per Theorem 1. We clarified this relationship in Section 5, and add results for GIN+IGEL in Table 3 from new experiments.
>
> A key difference of ELENE is that GSNs [2] require a choice of substructure to count. Although substructures like paths or cycles can be computed efficiently, in general, preprocessing a $k$-node substructure has an exponential cost in $k$. Other difference is that structural counts are only introduced as-is during learning, rather than embedded with node / edge attributes, as in ELENE-L. Table 3 shows GSN underperforms GIN+ELENE on ZINC and MolHIV. We have added this in our revised version.
>
> > Can ELENE tackle graphs with >10^5 nodes? ELENE-L (ED) can hardly be employed on real-world datasets. Possible by adding graph sampling techniques?
>
> The limit of 10^4.5 nodes in our experiments is caused what our hardware (the CPU memory in our experiments is 96GB) and implementation can handle. Our implementation uses a sparse adjacency matrix which becomes dense when computing sub-graph distance encodings to leverage vectorization. Scaling to more than 10^5 nodes is possible with more CPU-memory or by changing the implementation. The latter can be done by computing encodings through parallel BFS, resulting in a slower algorithm of the same order of complexity. We have modified the revised paper with this clarification.
>
> The key takeaway is that edge-level information at the node level (i.e. ELENE (ND)) during learning scales well to large graphs, as GPU memory consumption in Fig. 7 shows. A sampling version of ELENE is also a good suggestion for improvements in this regard.
>
> > ELENE improves few baselines. In Table 3, ELENE achieves on-par perf. on two dataset, weaker on 3.
>
> Computing ELENE encodings is an efficient preprocessing step that can be done once, and is cheaper than training/inference on sub-graph GNN baselines. As noted above, compared to count-based methods like IGEL or GSNs, we find that GIN+ELENE outperforms them.
>
> > Evaluation with related baselines [1-2] or other real-world benchmarks. Expressivity vs. real-world settings needs demonstration.
>
> Note Table 3 shows results from GSN [2] on ZINC and MolHIV, and in both cases, GIN+ELENE outperforms previous results from GSN. We add new results for GIN+IGEL on three real-world Datasets (ZINC, PATTERN and MolHIV) and find that GIN+ELENE matches or outperforms their performance.
>
> A main finding of our work is that embedding edge-level information during learning is valuable, as it provides node / edge-level sub-graph information, improving expressivity (ELENE-L (ED)) or empirical performance (ELENE-L (ND)) as a function of compute cost. The link between k-WL expressivity and empirical performance is still unclear. We believe it is important for future work to explore if 3-WL expressivity is a meaningful measure of model capabilities in practical applications.
>
> > Unfair to compare GIN+ELENE and GIN-AK as param. compl. differs.
>
> We emphasize the noted comparison as GIN+ELENE yields statistically comparable or better results than GIN-AK at a significantly lower memory cost in four of our five real-world tasks. If anything, we believe that this comparison favors GIN-AK, since GIN-AK has more parameters.
>
> > ELENE-L (ND) has ~ memory usage to GIN-AK+ while ELENE-L (ED) uses more memory than GIN-AK+.
>
> In Table 4, ELENE-L (ND) uses less memory vs. GIN-AK^+ and outperforms GIN-AK in all 5 datasets less lower memory. It matches GIN-AK^+ in ZINC, MolHIV and MolPCBA, with 99.95% of the performance in PATTERN and memory costs between 1.46x (MolHIV) and 3.40x (PATTERN) lower.
>
> ELENE-L (ED) adds edge-level learning, which is more expressive than ELENE-L (ND) (theoretically in Section 5 and Table 1, on SR25). As noted, increased expressivity does not seem to directly translate in performance on real-world graphs.
>
> We believe that our incremental ablation study (encodings, node-level learning, node and edge-level learning) helps analyze memory usage as a function of where edge-level ego-network signals are introduced and how.
>
> > [2] is not correctly cited in journal.
>
> We correct [2] to refer to the published journal.
>
> > What is the difference between GIN-AK and GIN-AK+?
>
> GIN-AK^{+} is a variant of GIN-AK from [3] adding context encodings and distance-to-centroid embeddings that is more performant but costlier than GIN-AK.
>
> [3] From Stars to Subgraphs: Uplifting Any GNN with Local Structure Awareness, ICLR 2022.

---

> > ### Comment · Reviewer_ddSg · 2024-03-14
> > **Thank you for the response**
> >
> > Thank you for the reply. The authors have addressed most of my concerns and I have no further questions.

---

### Author Response · Authors · 2024-03-03
**Message to Reviewers**

We want to thank the reviewers again for their helpful feedback. We have uploaded a new version of the manuscript with the following changes:

- Pg. 1.: Removed references to robustness as we do not explore this topic in the remainder of the manuscript.
- Pg. 5.: Introduced citations in order of appearance in Section 3.3. and moved the sparse ELENE vector specification into Eq. 2. to highlight its definition.
- Pg. 6.: Highlighted that $\gamma$ is a learnable parameter.
- Pg. 8.: Introduced additional references to OSAN, Path-GNNs, and clarified the relationship of ELENE with IGEL, GSNs and ESC-GNN.
- Pg. 9.: Introduced references to learning-based sub-graph selection methods like MAG-GNN and Policy-Learn, as well as PathGCNs as an example of perturbation methods based on signals derived from random walks.
- Pg. 10.: Clarified the relationship of ELENE with IGEL in the expressivity proofs by highlighting that sparse IGEL vectors contain a strict subset of the information in ELENE encodings.
- Pg. 12.: Introduced GIN-AK and GIN-AK$^{+}$ in the set of models that we compare against on the memory scalability benchmark (D).
- Pg. 13.: Clarified the maximum memory consumption and number of CPU cores per job.
- Pg. 14.: Fixed a typo in “Given increased memory and computation costs and the weaker performance of Elene-L (ED)”, where “performance” was missing.
- Pg. 15.: Introduced GIN+IGEL results for ZINC, MolHIV and PATTERN in Table 3 and adjusted text sizes of literature results for better readability, as well as corrected typos. Clarified that Table 4 discusses time and memory performance. Condensed the summary of results and included discussion about GIN+IGEL. Added a reference to the environment described in Section 7.1 in which Table 4 is collected in the caption.
- Pg. 16.: Modified Figure 7. to include GIN-AK and GIN-AK$^{+}$ and further validate our memory consumption claims. Introduced references to extended results with different graph density patterns in Appendix B.2. Clarified CPU memory consumption and scalability concerns with graphs using more than 10^5 nodes. Highlighted that our additional results show ELENE (ND) outperforms GIN-AK and GIN-AK^{+}, consistent with Real World Datasets from Table 4.
- Pg. 17.: Added GIN-AK and GIN-AK$^{+}$ summarized results for the memory scalability benchmark to incorporate our updates in Section 7.5.
- Pg. 25.: Extended the "Memory Scalability" section of Appendix B.2. by (a) updating Figures 10 and 11 with results including GIN-AK and GIN-AK$^{+}$, (b) introducing Figure 12 where the value of $N=1000$ is kept constant and the degree of $k$-regular graphs is evaluated with $k \in \{6, 12, 18, 24, 30, 36\}$, (c) introducing Figure 13 where we explore an irregular density case using random graphs generated with the Barabási-Albert preferential attachment model and increasing number of minimum edges per node ($m$).

---

> ### Comment · Reviewer_suwS · 2024-03-11
> **Thanks**
>
> I thank the authors for the detailed response. I am satisfied with the proposed changes and clarifications in the revised manuscript.

---

### Decision · Action_Editor_8ydJ · 2024-04-04

**Recommendation:** Accept as is

**Comment:**

The paper was reviewed by three expert reviewers. All reviewers recommended acceptance of the paper and I agree with them. The reviewers initially raised concerns about the scalability of the proposed approach, the lack of baselines, the limited number of datasets and types of graphs. The reviewers also complained about missing related work. Those concerns were addressed by the authors in the revision and the quality of the paper improved significantly. I thus think that the paper is now ready for publication.

**Audience:**

The topic and findings of the paper are of interest to several individuals in TMLR's audience, mainly individuals working in the field of graph neural networks and their expressive power.

**Claims And Evidence:**

This work proposes to inject structural features into graph neural networks. Those features capture information at the edge-level, including signals contained in the ego-networks of adjacent nodes. The authors claim that their approach achieves comparable performance with state-of-the-art learning methods. This claim is supported by the reported empirical results. The authors also claim that the proposed approach can distinguish strongly regular graphs, and this claim is also supported by both a proof and empirical results.